# Training Generative Adversarial Networks with Limited Data

**Tero Karras**
NVIDIA
tkarras@nvidia.com

**Miika Aittala**
NVIDIA
maittala@nvidia.com

**Janne Hellsten**
NVIDIA
jhellsten@nvidia.com

**Samuli Laine**
NVIDIA
slaine@nvidia.com

**Jaakko Lehtinen**
NVIDIA and Aalto University
jlehtinen@nvidia.com

**Timo Aila**
NVIDIA
taila@nvidia.com

## Abstract

Training generative adversarial networks (GAN) using too little data typically leads to discriminator overfitting, causing training to diverge. We propose an adaptive discriminator augmentation mechanism that significantly stabilizes training in limited data regimes. The approach does not require changes to loss functions or network architectures, and is applicable both when training from scratch and when fine-tuning an existing GAN on another dataset. We demonstrate, on several datasets, that good results are now possible using only a few thousand training images, often matching StyleGAN2 results with an order of magnitude fewer images. We expect this to open up new application domains for GANs. We also find that the widely used CIFAR-10 is, in fact, a limited data benchmark, and improve the record FID from 5.59 to 2.42.

## 1 Introduction

The increasingly impressive results of generative adversarial networks (GAN) [12, 25, 24, 5, 17, 18, 19] are fueled by the seemingly unlimited supply of images available online. Still, it remains challenging to collect a large enough set of images for a specific application that places constraints on subject type, image quality, geographical location, time period, privacy, copyright status, etc. The difficulties are further exacerbated in applications that require the capture of a new, custom dataset: acquiring, processing, and distributing the $\sim 10^5 - 10^6$ images required to train a modern high-quality, high-resolution GAN is a costly undertaking. This curbs the increasing use of generative models in fields such as medicine [38]. A significant reduction in the number of images required therefore has the potential to considerably help many applications.

The key problem with small datasets is that the discriminator overfits to the training examples; its feedback to the generator becomes meaningless and training starts to diverge [2, 39]. In almost all areas of deep learning [32], *dataset augmentation* is the standard solution against overfitting. For example, training an image classifier under rotation, noise, etc., leads to increasing invariance to these semantics-preserving distortions — a highly desirable quality in a classifier [15, 8, 9]. In contrast, a GAN trained under similar dataset augmentations learns to generate the augmented distribution [40, 43]. In general, such "leaking" of augmentations to the generated samples is highly undesirable. For example, a noise augmentation leads to noisy results, even if there is none in the dataset.

In this paper, we demonstrate how to use a wide range of augmentations to prevent the discriminator from overfitting, while ensuring that none of the augmentations leak to the generated images. We start by presenting a comprehensive analysis of the conditions that prevent the augmentations from leaking. We then design a diverse set of augmentations, and an adaptive control scheme that enables

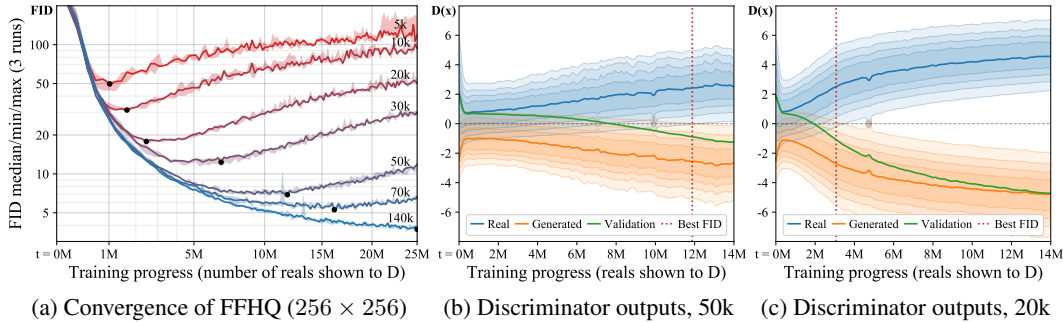

(a) Convergence of FFHQ ($256 \times 256$)  (b) Discriminator outputs, 50k  (c) Discriminator outputs, 20k

Figure 1: (a) Convergence with different training set sizes. "140k" means that we amplified the 70k dataset by $2\times$ through $x$-flips; we do not use data amplification in any other case. (b,c) Evolution of discriminator outputs during training. Each vertical slice shows a histogram of $D(x)$, i.e., raw logits.

the same approach to be used regardless of the amount of training data, properties of the dataset, or the exact training setup (e.g., training from scratch or transfer learning [26, 36, 37, 27]).

We demonstrate, on several datasets, that good results are now possible using only a few thousand images, often matching StyleGAN2 results with an order of magnitude fewer images. Furthermore, we show that the popular CIFAR-10 benchmark suffers from limited data and achieve a new record Fréchet inception distance (FID) [16] of 2.42, significantly improving over the current state of the art of 5.59 [42]. We also present METFACES, a high-quality benchmark dataset for limited data scenarios. Our implementation and models are available at `https://github.com/NVlabs/stylegan2-ada`

## 2  Overfitting in GANs

We start by studying how the quantity of available training data affects GAN training. We approach this by artificially subsetting larger datasets (FFHQ and LSUN CAT) and observing the resulting dynamics. For our baseline, we considered StyleGAN2 [19] and BigGAN [5, 30]. Based on initial testing, we settled on StyleGAN2 because it provided more predictable results with significantly lower variance between training runs (see Appendix A in the Supplement). For each run, we randomize the subset of training data, order of training samples, and network initialization. To facilitate extensive sweeps over dataset sizes and hyperparameters, we use a downscaled $256 \times 256$ version of FFHQ and a lighter-weight configuration that reaches the same quality as the official StyleGAN2 config F for this dataset, but runs $4.6\times$ faster on NVIDIA DGX-1.[1] We measure quality by computing FID between 50k generated images and all available training images, as recommended by Heusel et al. [16], regardless of the subset actually used for training.

Figure 1a shows our baseline results for different subsets of FFHQ. Training starts the same way in each case, but eventually the progress stops and FID starts to rise. The less training data there is, the earlier this happens. Figure 1b,c shows the discriminator output distributions for real and generated images during training. The distributions overlap initially but keep drifting apart as the discriminator becomes more and more confident, and the point where FID starts to deteriorate is consistent with the loss of sufficient overlap between distributions. This is a strong indication of overfitting, evidenced further by the drop in accuracy measured for a separate validation set. We propose a way to tackle this problem by employing versatile augmentations that prevent the discriminator from becoming overly confident.

### 2.1  Stochastic discriminator augmentation

By definition, any augmentation that is applied to the training dataset will get inherited to the generated images [12]. Zhao et al. [43] recently proposed balanced consistency regularization (bCR) as a solution that is not supposed to leak augmentations to the generated images. Consistency regularization states that two sets of augmentations, applied to the same input image, should yield the same output [28, 23]. Zhao et al. add consistency regularization terms for the discriminator loss, and

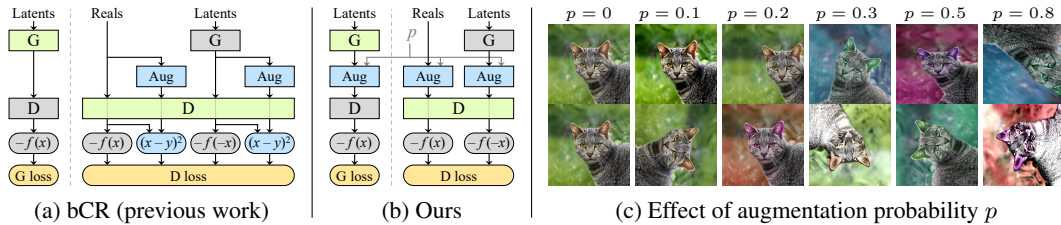

(a) bCR (previous work)  (b) Ours  (c) Effect of augmentation probability $p$

Figure 2: (a,b) Flowcharts for balanced consistency regularization (bCR) [43] and our stochastic discriminator augmentations. The blue elements highlight operations related to augmentations, while the rest implement standard GAN training with generator $G$ and discriminator $D$ [12]. The orange elements indicate the loss function and the green boxes mark the network being trained. We use the non-saturating logistic loss [12] $f(x) = \log(\text{sigmoid}(x))$. (c) We apply a diverse set of augmentations to every image that the discriminator sees, controlled by an augmentation probability $p$.

enforce discriminator consistency for both real and generated images, whereas no augmentations or consistency loss terms are applied when training the generator (Figure 2a). As such, their approach effectively strives to generalize the discriminator by making it blind to the augmentations used in the CR term. However, meeting this goal opens the door for leaking augmentations, because the generator will be free to produce images containing them without any penalty. In Section 4, we show experimentally that bCR indeed suffers from this problem, and thus its effects are fundamentally similar to dataset augmentation.

Our solution is similar to bCR in that we also apply a set of augmentations to all images shown to the discriminator. However, instead of adding separate CR loss terms, we evaluate the discriminator *only* using augmented images, and do this also when training the generator (Figure 2b). This approach that we call *stochastic discriminator augmentation* is therefore very straightforward. Yet, this possibility has received little attention, possibly because at first glance it is not obvious if it even works: if the discriminator never sees what the training images really look like, it is not clear if it can guide the generator properly (Figure 2c). We will therefore first investigate the conditions under which this approach will not leak an augmentation to the generated images, and then build a full pipeline out of such transformations.

## 2.2 Designing augmentations that do not leak

Discriminator augmentation corresponds to putting distorting, perhaps even destructive goggles on the discriminator, and asking the generator to produce samples that cannot be distinguished from the training set when viewed through the goggles. Bora et al. [4] consider a similar problem in training GANs under corrupted measurements, and show that the training *implicitly* undoes the corruptions and finds the correct distribution, as long as the corruption process is represented by an invertible transformation of probability distributions over the data space. We call such augmentation operators *non-leaking*.

The power of these invertible transformations is that they allow conclusions about the equality or inequality of the underlying sets to be drawn by observing only the augmented sets. It is crucial to understand that this does *not* mean that augmentations performed on individual images would need to be undoable. For instance, an augmentation as extreme as setting the input image to zero 90% of the time is invertible in the probability distribution sense: it would be easy, even for a human, to reason about the original distribution by ignoring black images until only 10% of the images remain. On the other hand, random rotations chosen uniformly from $\{0°, 90°, 180°, 270°\}$ are not invertible: it is impossible to discern differences among the orientations after the augmentation.

The situation changes if this rotation is only executed at a probability $p < 1$: this increases the relative occurrence of $0°$, and now the augmented distributions can match only if the generated images have correct orientation. Similarly, many other stochastic augmentations can be designed to be non-leaking on the condition that they are skipped with a non-zero probability. Appendix C shows that this can be made to hold for a large class of widely used augmentations, including deterministic mappings (e.g., basis transformations), additive noise, transformation groups (e.g, image or color space rotations, flips and scaling), and projections (e.g., cutout [10]). Furthermore, composing non-leaking augmentations in a fixed order yields an overall non-leaking augmentation.

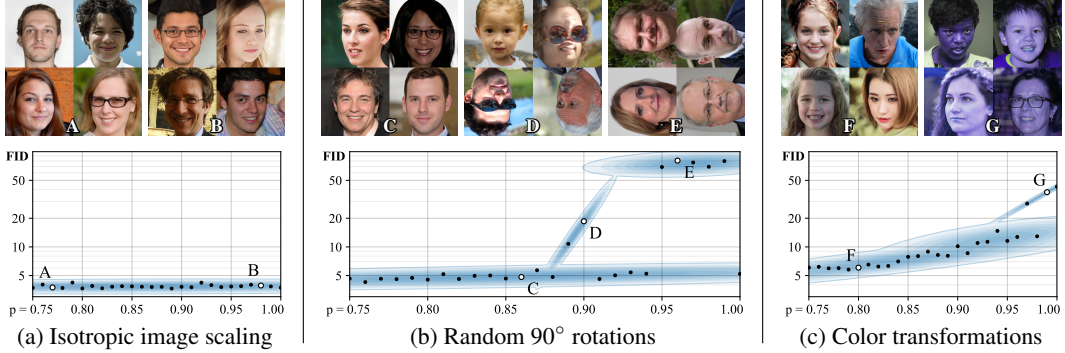

| (a) Isotropic image scaling | (b) Random 90° rotations | (c) Color transformations |

Figure 3: Leaking behavior of three example augmentations, shown as FID w.r.t. the probability of executing the augmentation. Each dot represents a complete training run, and the blue Gaussian mixture is a visualization aid. The top row shows generated example images from selected training runs, indicated by uppercase letters in the plots.

In Figure 3 we validate our analysis by three practical examples. Isotropic scaling with log-normal distribution is an example of an inherently safe augmentation that does not leak regardless of the value of $p$ (Figure 3a). However, the aforementioned rotation by a random multiple of 90° must be skipped at least part of the time (Figure 3b). When $p$ is too high, the generator cannot know which way the generated images should face and ends up picking one of the possibilities at random. As could be expected, the problem does not occur exclusively in the limiting case of $p = 1$. In practice, the training setup is poorly conditioned for nearby values as well due to finite sampling, finite representational power of the networks, inductive bias, and training dynamics. When $p$ remains below $\sim 0.85$, the generated images are always oriented correctly. Between these regions, the generator sometimes picks a wrong orientation initially, and then partially drifts towards the correct distribution. The same observations hold for a sequence of continuous color augmentations (Figure 3c). This experiment suggests that as long as $p$ remains below $0.8$, leaks are unlikely to happen in practice.

## 2.3 Our augmentation pipeline

We start from the assumption that a maximally diverse set of augmentations is beneficial, given the success of RandAugment [9] in image classification tasks. We consider a pipeline of 18 transformations that are grouped into 6 categories: pixel blitting ($x$-flips, 90° rotations, integer translation), more general geometric transformations, color transforms, image-space filtering, additive noise [33], and cutout [10]. Details of the individual augmentations are given in Appendix B. Note that we execute augmentations also when training the generator (Figure 2b), which requires the augmentations to be differentiable. We achieve this by implementing them using standard differentiable primitives offered by the deep learning framework.

During training, we process each image shown to the discriminator using a pre-defined set of transformations in a fixed order. The strength of augmentations is controlled by the scalar $p \in [0, 1]$, so that each transformation is applied with probability $p$ or skipped with probability $1 - p$. We always use the same value of $p$ for all transformations. The randomization is done separately for each augmentation and for each image in a minibatch. Given that there are many augmentations in the pipeline, even fairly small values of $p$ make it very unlikely that the discriminator sees a clean image (Figure 2c). Nonetheless, the generator is guided to produce only clean images as long as $p$ remains below the practical safety limit.

In Figure 4 we study the effectiveness of stochastic discriminator augmentation by performing exhaustive sweeps over $p$ for different augmentation categories and dataset sizes. We observe that it can improve the results significantly in many cases. However, the optimal augmentation strength depends heavily on the amount of training data, and not all augmentation categories are equally useful in practice. With a 2k training set, the vast majority of the benefit came from pixel blitting and geometric transforms. Color transforms were modestly beneficial, while image-space filtering, noise, and cutout were not particularly useful. In this case, the best results were obtained using strong augmentations. The curves also indicate some of the augmentations becoming leaky when $p \to 1$. With a 10k training set, the higher values of $p$ were less helpful, and with 140k the situation was

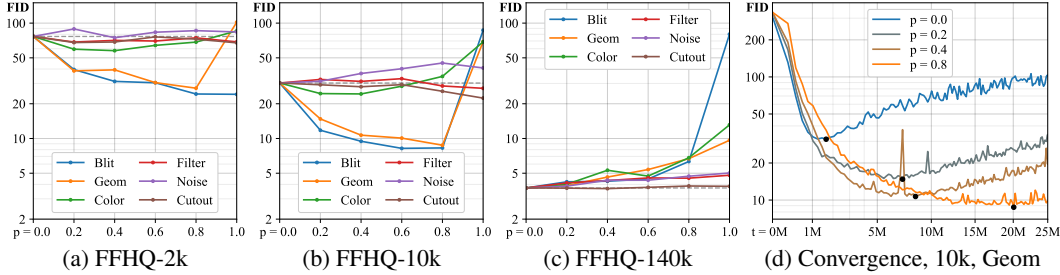

Figure 4: (a-c) Impact of $p$ for different augmentation categories and dataset sizes. The dashed gray line indicates baseline FID without augmentations. (d) Convergence curves for selected values of $p$ using geometric augmentations with 10k training images.

markedly different: all augmentations were harmful. Based on these results, we choose to use only pixel blitting, geometric, and color transforms for the rest of our tests. Figure 4d shows that while stronger augmentations reduce overfitting, they also slow down the convergence.

In practice, the sensitivity to dataset size mandates a costly grid search, and even so, relying on any fixed $p$ may not be the best choice. Next, we address these concerns by making the process adaptive.

## 3   Adaptive discriminator augmentation

Ideally, we would like to avoid manual tuning of the augmentation strength and instead control it dynamically based on the degree of overfitting. Figure 1 suggests a few possible approaches for this. The standard way of quantifying overfitting is to use a separate validation set and observe its behavior relative to the training set. From the figure we see that when overfitting kicks in, the validation set starts behaving increasingly like the generated images. This is a quantifiable effect, albeit with the drawback of requiring a separate validation set when training data may already be in short supply. We can also see that with the non-saturating loss [12] used by StyleGAN2, the discriminator outputs for real and generated images diverge symmetrically around zero as the situation gets worse. This divergence can be quantified without a separate validation set.

Let us denote the discriminator outputs by $D_{\text{train}}$, $D_{\text{validation}}$, and $D_{\text{generated}}$ for the training set, validation set, and generated images, respectively, and their mean over $N$ consecutive minibatches by $\mathbb{E}[\cdot]$. In practice we use $N = 4$, which corresponds to $4 \times 64 = 256$ images. We can now turn our observations about Figure 1 into two plausible overfitting heuristics:

$$r_v = \frac{\mathbb{E}[D_{\text{train}}] - \mathbb{E}[D_{\text{validation}}]}{\mathbb{E}[D_{\text{train}}] - \mathbb{E}[D_{\text{generated}}]} \qquad\qquad r_t = \mathbb{E}[\text{sign}(D_{\text{train}})] \qquad (1)$$

For both heuristics, $r = 0$ means no overfitting and $r = 1$ indicates complete overfitting, and our goal is to adjust the augmentation probability $p$ so that the chosen heuristic matches a suitable target value. The first heuristic, $r_v$, expresses the output for a validation set relative to the training set and generated images. Since it assumes the existence of a separate validation set, we include it mainly as a comparison method. The second heuristic, $r_t$, estimates the portion of the training set that gets positive discriminator outputs. We have found this to be far less sensitive to the chosen target value and other hyperparameters than the obvious alternative of looking at $\mathbb{E}[D_{\text{train}}]$ directly.

We control the augmentation strength $p$ as follows. We initialize $p$ to zero and adjust its value once every four minibatches[2] based on the chosen overfitting heuristic. If the heuristic indicates too much/little overfitting, we counter by incrementing/decrementing $p$ by a fixed amount. We set the adjustment size so that $p$ can rise from 0 to 1 sufficiently quickly, e.g., in 500k images. After every step we clamp $p$ from below to 0. We call this variant *adaptive discriminator augmentation* (ADA).

In Figure 5a,b we measure how the target value affects the quality obtainable using these heuristics. We observe that $r_v$ and $r_t$ are both effective in preventing overfitting, and that they both improve the results over the best fixed $p$ found using grid search. We choose to use the more realistic $r_t$ heuristic in all subsequent tests, with 0.6 as the target value. Figure 5c shows the resulting $p$ over time. With a

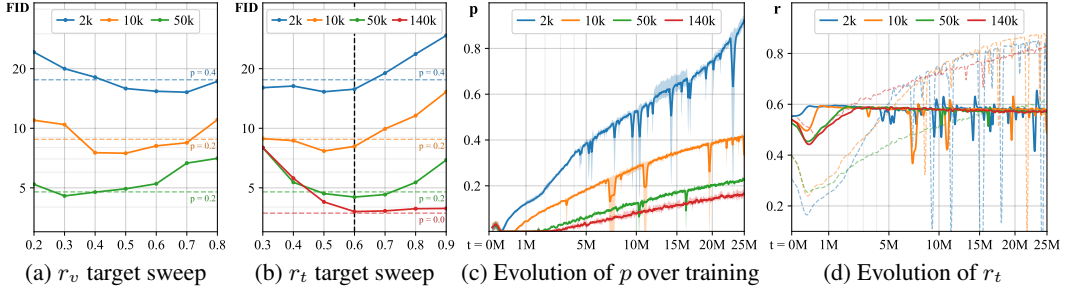

(a) $r_v$ target sweep     (b) $r_t$ target sweep     (c) Evolution of $p$ over training     (d) Evolution of $r_t$

Figure 5: Behavior of our adaptive augmentation strength heuristics in FFHQ. (a,b) FID for different training set sizes as a function of the target value for $r_v$ and $r_t$. The dashed horizontal lines indicate the best fixed augmentation probability $p$ found using grid search, and the dashed vertical line marks the target value we will use in subsequent tests. (c) Evolution of $p$ over the course of training using heuristic $r_t$. (d) Evolution of $r_t$ values over training. Dashes correspond to the fixed $p$ values in (b).

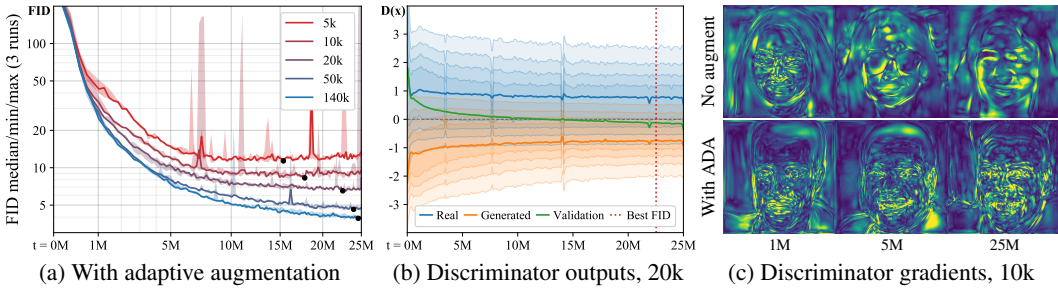

(a) With adaptive augmentation     (b) Discriminator outputs, 20k     (c) Discriminator gradients, 10k

Figure 6: (a) Training curves for FFHQ with different training set sizes using adaptive augmentation. (b) The supports of real and generated images continue to overlap. (c) Example magnitudes of the gradients the generator receives from the discriminator as the training progresses.

2k training set, augmentations were applied almost always towards the end. This exceeds the practical safety limit after which some augmentations become leaky, indicating that the augmentations were not powerful enough. Indeed, FID started deteriorating after $p \approx 0.5$ in this extreme case. Figure 5d shows the evolution of $r_t$ with adaptive vs fixed $p$, showing that a fixed $p$ tends to be too strong in the beginning and too weak towards the end.

Figure 6 repeats the setup from Figure 1 using ADA. Convergence is now achieved regardless of the training set size and overfitting no longer occurs. Without augmentations, the gradients the generator receives from the discriminator become very simplistic over time — the discriminator starts to pay attention to only a handful of features, and the generator is free to create otherwise non-sensical images. With ADA, the gradient field stays much more detailed which prevents such deterioration. In an interesting parallel, it has been shown that loss functions can be made significantly more robust in regression settings by using similar image augmentation ensembles [21].

## 4 Evaluation

We start by testing our method against a number of alternatives in FFHQ and LSUN CAT, first in a setting where a GAN is trained from scratch, then by applying transfer learning on a pre-trained GAN. We conclude with results for several smaller datasets.

### 4.1 Training from scratch

Figure 7 shows our results in FFHQ and LSUN CAT across training set sizes, demonstrating that our adaptive discriminator augmentation (ADA) improves FIDs substantially in limited data scenarios. We also show results for balanced consistency regularization (bCR) [43], which has not been studied in the context of limited data before. We find that bCR can be highly effective when the lack of data is not too severe, but also that its set of augmentations leaks to the generated images. In this example, we used only $xy$-translations by integer offsets for bCR, and Figure 7d shows that the generated images get jittered as a result. This means that bCR is essentially a dataset augmentation and needs

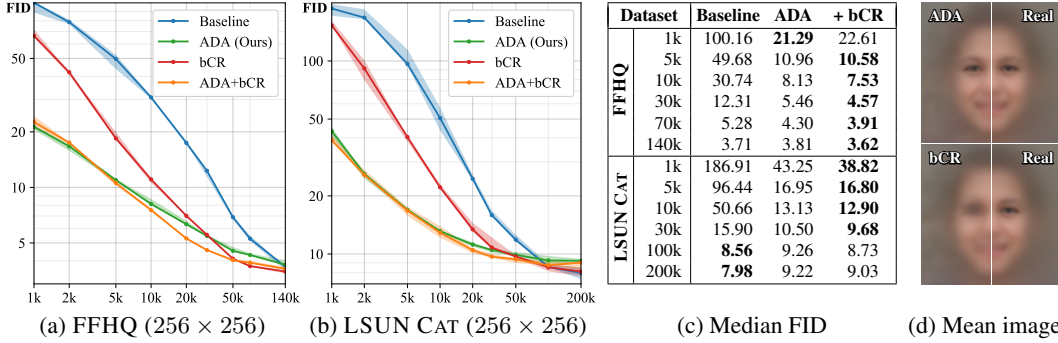

Figure 7: (a-c) FID as a function of training set size, reported as median/min/max over 3 training runs. (d) Average of 10k random images generated using the networks trained with 5k subset of FFHQ. ADA matches the average of real data, whereas the $xy$-translation augmentation in bCR [43] has leaked to the generated images, significantly blurring the average image.

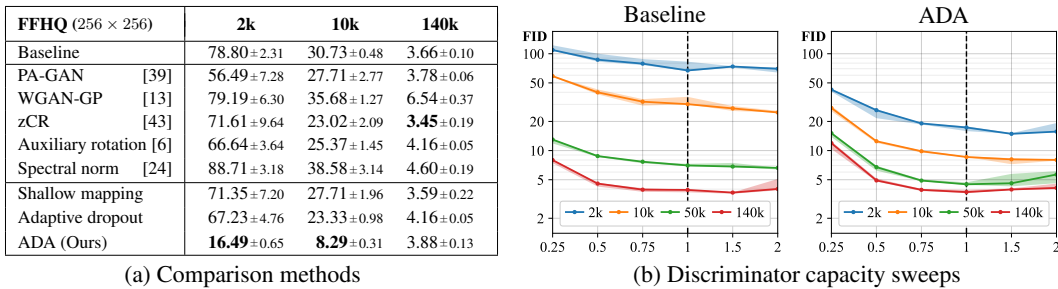

Figure 8: (a) We report the mean and standard deviation for each comparison method, calculated over 3 training runs. (b) FID as a function of discriminator capacity, reported as median/min/max over 3 training runs. We scale the number of feature maps uniformly across all layers by a given factor ($x$-axis). The baseline configuration (no scaling) is indicated by the dashed vertical line.

to be limited to symmetries that actually benefit the training data, e.g., $x$-flip is often acceptable but $y$-flip only rarely. Meanwhile, with ADA the augmentations do not leak, and thus the same diverse set of augmentations can be safely used in all datasets. We also find that the benefits for ADA and bCR are largely additive. We combine ADA and bCR so that ADA is first applied to the input image (real or generated), and bCR then creates another version of this image using *its own set of augmentations*. Qualitative results are shown in Appendix A.

In Figure 8a we further compare our adaptive augmentation against a wider set of alternatives: PA-GAN [39], WGAN-GP [13], zCR [43], auxiliary rotations [6], and spectral normalization [24]. We also try modifying our baseline to use a shallower mapping network, which can be trained with less data, borrowing intuition from DeLiGAN [14]. Finally, we try replacing our augmentations with multiplicative dropout [34], whose per-layer strength is driven by our adaptation algorithm. We spent considerable effort tuning the parameters of all these methods, see Appendix D. We can see that ADA gave significantly better results than the alternatives. While PA-GAN is somewhat similar to our method, its checksum task was not strong enough to prevent overfitting in our tests. Figure 8b shows that reducing the discriminator capacity is generally harmful and does not prevent overfitting.

## 4.2 Transfer learning

Transfer learning reduces the training data requirements by starting from a model trained using some other dataset, instead of a random initialization. Several authors have explored this in the context of GANs [36, 37, 27], and Mo et al. [26] recently showed strong results by freezing the highest-resolution layers of the discriminator during transfer (Freeze-D).

We explore several transfer learning setups in Figure 9, using the best Freeze-D configuration found for each case with grid search. Transfer learning gives significantly better results than from-scratch training, and its success seems to depend primarily on the diversity of the source dataset, instead of

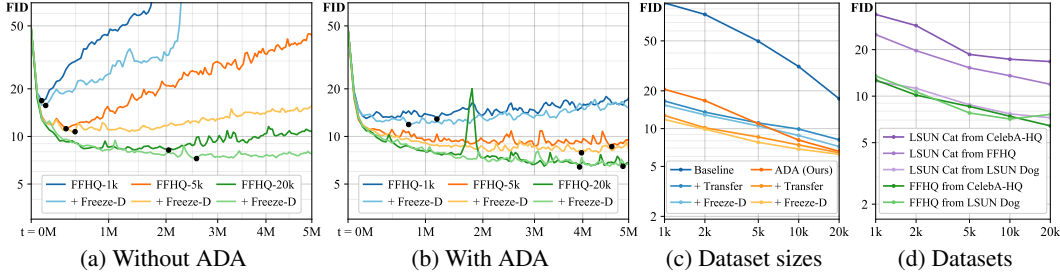

| (a) Without ADA | (b) With ADA | (c) Dataset sizes | (d) Datasets |

Figure 9: Transfer learning FFHQ starting from a pre-trained CELEBA-HQ model, both $256 \times 256$. (a) Training convergence for our baseline method and Freeze-D [26]. (b) The same configurations with ADA. (c) FIDs as a function of dataset size. (d) Effect of source and target datasets.

| METFACES (new dataset) | BRECAHAD | AFHQ CAT, DOG, WILD ($512^2$) | CIFAR-10 |
| 1336 img, $1024^2$, transfer learning from FFHQ | 1944 img, $512^2$ | 5153 img    4739 img    4738 img | 50k, 10 cls, $32^2$ |

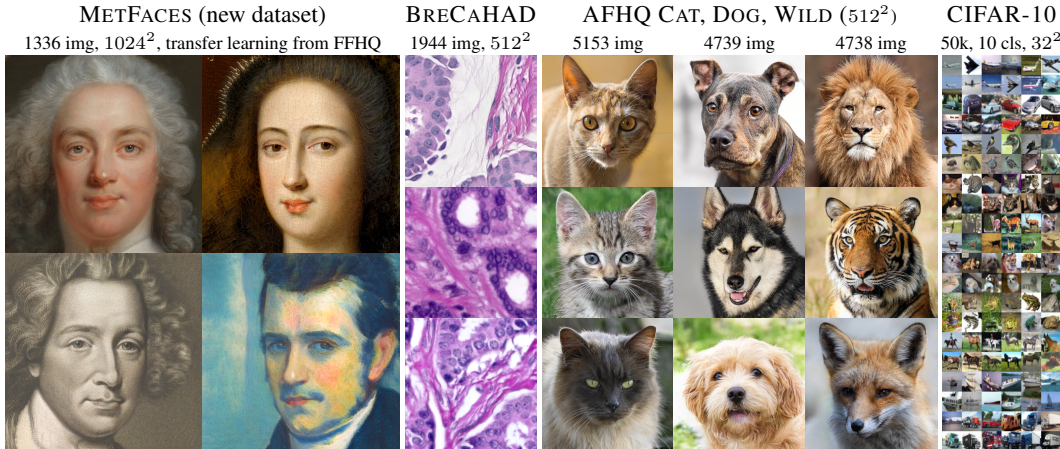

Figure 10: Example generated images for several datasets with limited amount of training data, trained using ADA. We use transfer learning with METFACES and train other datasets from scratch. See Appendix A for uncurated results and real images, and Appendix D for our training configurations.

the similarity between subjects. For example, FFHQ (human faces) can be trained equally well from CELEBA-HQ (human faces, low diversity) or LSUN DOG (more diverse). LSUN CAT, however, can only be trained from LSUN DOG, which has comparable diversity, but not from the less diverse datasets. With small target dataset sizes, our baseline achieves reasonable FID quickly, but the progress soon reverts as training continues. ADA is again able to prevent the divergence almost completely. Freeze-D provides a small but reliable improvement when used together with ADA but is not able to prevent the divergence on its own.

## 4.3 Small datasets

We tried our method with several datasets that consist of a limited number of training images (Figure 10). METFACES is our new dataset of 1336 high-quality faces extracted from the collection of Metropolitan Museum of Art (`https://metmuseum.github.io/`). BRECAHAD [1] consists of only 162 breast cancer histopathology images ($1360 \times 1024$); we reorganized these into 1944 partially overlapping crops of $512^2$. Animal faces (AFHQ) [7] includes ~5k closeups per category for dogs, cats, and wild life; we treated these as three separate datasets and trained a separate network for each of them. CIFAR-10 includes 50k tiny images in 10 categories [22].

Figure 11 reveals that FID is not an ideal metric for small datasets, because it becomes dominated by the inherent bias when the number of real images is insufficient. We find that kernel inception distance (KID) [3] — that is unbiased by design — is more descriptive in practice and see that ADA provides a dramatic improvement over baseline StyleGAN2. This is especially true when training from scratch, but transfer learning also benefits from ADA. In the widely used CIFAR-10 benchmark, we improve the SOTA FID from 5.59 to 2.42 and inception score (IS) [29] from 9.58 to 10.24 in the class-conditional setting (Figure 11b). This large improvement portrays CIFAR-10 as a limited data benchmark. We also note that CIFAR-specific architecture tuning had a significant effect.

|  |  | Scratch | | Transfer | + Freeze-D |
|---|---|---|---|---|---|
| Dataset | Method | FID | KID $\times 10^3$ | KID $\times 10^3$ | KID $\times 10^3$ |
| METFACES | Baseline | 57.26 | 35.66 | 3.16 | 2.05 |
|  | ADA | **18.22** | **2.41** | **0.81** | **1.33** |
| BRECAHAD | Baseline | 97.72 | 89.76 | 18.07 | 6.94 |
|  | ADA | **15.71** | **2.88** | **3.36** | **1.91** |
| AFHQ CAT | Baseline | 5.13 | 1.54 | 1.09 | 1.00 |
|  | ADA | **3.55** | **0.66** | **0.44** | **0.35** |
| AFHQ DOG | Baseline | 19.37 | 9.62 | 4.63 | 2.80 |
|  | ADA | **7.40** | **1.16** | **1.40** | **1.12** |
| AFHQ WILD | Baseline | 3.48 | 0.77 | 0.31 | **0.12** |
|  | ADA | **3.05** | **0.45** | **0.15** | 0.14 |

(a) Small datasets

| Method | Unconditional | | Conditional | |
|---|---|---|---|---|
|  | FID ↓ | IS ↑ | FID ↓ | IS ↑ |
| ProGAN [17] | 15.52 | $8.56 \pm 0.06$ | – | – |
| AutoGAN [11] | 12.42 | $8.55 \pm 0.10$ | – | – |
| BigGAN [5] | – | – | 14.73 | 9.22 |
| + Tuning [20] | – | – | 8.47 | $9.07 \pm 0.13$ |
| MultiHinge [20] | – | – | 6.40 | $9.58 \pm 0.09$ |
| FQ-GAN [42] | – | – | $5.59 \pm 0.12$ | 8.48 |
| Baseline | $8.32 \pm 0.09$ | $9.21 \pm 0.09$ | $6.96 \pm 0.41$ | $9.53 \pm 0.06$ |
| + ADA (Ours) | $5.33 \pm 0.35$ | **$10.02 \pm 0.07$** | $3.49 \pm 0.17$ | **$10.24 \pm 0.07$** |
| + Tuning (Ours) | **$2.92 \pm 0.05$** | $9.83 \pm 0.04$ | **$2.42 \pm 0.04$** | $10.14 \pm 0.09$ |

(b) CIFAR-10

Figure 11: (a) Several small datasets trained with StyleGAN2 baseline (config F) and ADA, from scratch and using transfer learning. We used FFHQ-140K with matching resolution as a starting point for all transfers. We report the best KID, and compute FID using the same snapshot. (c) Mean and standard deviation for CIFAR-10, computed from the best scores of 5 training runs. For the comparison methods we report the average scores when available, and the single best score otherwise. The best IS and FID were searched separately [20], and often came from different snapshots. We computed the FID for Progressive GAN [17] using the publicly available pre-trained network.

## 5  Conclusions

We have shown that our adaptive discriminator augmentation reliably stabilizes training and vastly improves the result quality when training data is in short supply. Of course, augmentation is not a substitute for real data — one should always try to collect a large, high-quality set of training data first, and only then fill the gaps using augmentation. As future work, it would be worthwhile to search for the most effective set of augmentations, and to see if recently published techniques, such as the U-net discriminator [30] or multi-modal generator [31], could also help with limited data.

Enabling ADA has a negligible effect on the energy consumption of training a single model. As such, using it does not increase the cost of training models for practical use or developing methods that require large-scale exploration. For reference, Appendix E provides a breakdown of all computation that we performed related to this paper; the project consumed a total of 325 MWh of electricity, or 135 single-GPU years, the majority of which can be attributed to extensive comparisons and sweeps.

Interestingly, the core idea of discriminator augmentations was independently discovered by three other research groups in parallel work: Z. Zhao et al. [44], Tran et al. [35], and S. Zhao et al. [41]. We recommend these papers as they all offer a different set of intuition, experiments, and theoretical justifications. While two of these papers [44, 41] propose essentially the same augmentation mechanism as we do, they study the absence of leak artifacts only empirically. The third paper [35] presents a theoretical justification based on invertibility, but arrives at a different argument that leads to a more complex network architecture, along with significant restrictions on the set of possible augmentations. None of these works consider the possibility of tuning augmentation strength adaptively. Our experiments in Section 3 show that the optimal augmentation strength not only varies between datasets of different content and size, but also over the course of training — even an optimal set of fixed augmentation parameters is likely to leave performance on the table.

A direct comparison of results between the parallel works is difficult because the only dataset used in all papers is CIFAR-10. Regrettably, the other three papers compute FID using 10k generated images and 10k *validation* images (FID-10k), while we use follow the original recommendation of Heusel et al. [16] and use 50k generated images and all *training* images. Their FID-10k numbers are thus not comparable to the FIDs in Figure 11b. For this reason we also computed FID-10k for our method, obtaining $7.01 \pm 0.06$ for unconditional and $6.54 \pm 0.06$ for conditional. These compare favorably to parallel work's unconditional 9.89 [41] or 10.89 [35], and conditional 8.30 [44] or 8.49 [41]. It seems likely that some combination of the ideas from all four papers could further improve our results. For example, more diverse set of augmentations or contrastive regularization [44] might be worth testing.

**Acknowledgements**  We thank David Luebke for helpful comments; Tero Kuosmanen and Sabu Nadarajan for their support with compute infrastructure; and Edgar Schönfeld for guidance on setting up unconditional BigGAN. We did not receive external funding or additional revenues for this project.

## Broader impact

Data-driven generative modeling means learning a computational recipe for generating complicated data based purely on examples. This is a foundational problem in machine learning. In addition to their fundamental nature, generative models have several uses within applied machine learning research as priors, regularizers, and so on. In those roles, they advance the capabilities of computer vision and graphics algorithms for analyzing and synthesizing realistic imagery.

The methods presented in this work enable high-quality generative image models to be trained using significantly less data than required by existing approaches. It thereby primarily contributes to the deep technical question of how much data is enough for generative models to succeed in picking up the necessary commonalities and relationships in the data.

From an applied point of view, this work contributes to efficiency; it does not introduce fundamental new capabilities. Therefore, it seems likely that the advances here will not substantially affect the overall themes — surveillance, authenticity, privacy, etc. — in the active discussion on the broader impacts of computer vision and graphics.

Specifically, generative models' implications on image and video authenticity is a topic of active discussion. Most attention revolves around conditional models that allow semantic control and sometimes manipulation of existing images. Our algorithm does not offer direct controls for high-level attributes (e.g., identity, pose, expression of people) in the generated images, nor does it enable direct modification of existing images. However, over time and through the work of other researchers, our advances will likely lead to improvements in these types of models as well.

The contributions in this work make it easier to train high-quality generative models with custom sets of images. By this, we eliminate, or at least significantly lower, the barrier for applying GAN-type models in many applied fields of research. We hope and believe that this will accelerate progress in several such fields. For instance, modeling the space of possible appearance of biological specimens (tissues, tumors, etc.) is a growing field of research that appears to chronically suffer from limited high-quality data. Overall, generative models hold promise for increased understanding of the complex and hard-to-pinpoint relationships in many real-world phenomena; our work hopefully increases the breadth of phenomena that can be studied.

## Footnotes

[1]We use $2\times$ fewer feature maps, $2\times$ larger minibatch, mixed-precision training for layers at $\geq 32^2$, $\eta = 0.0025$, $\gamma = 1$, and exponential moving average half-life of 20k images for generator weights.

[2]This choice follows from StyleGAN2 training loop layout. The results are not sensitive to this parameter.

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
