[Supplementary Material]

# Supplemental Material:
# Training Generative Adversarial Networks with Limited Data

We present additional result images and comparisons in Appendix A. We then proceed to describe our augmentation pipeline in Appendix B and analyze the augmentations from a theoretical standpoint in Appendix C. Finally, we detail our training setup in Appendix D and present a detailed breakdown of our total energy consumption in Appendix E.

## A    Additional results

In Figures 12, 13, 14, 15, and 16, we show generated images for METFACES, BRECAHAD, and AFHQ CAT, DOG, WILD, respectively, along with real images from the respective training sets (Section 4.3 and Figure 11a). The images were selected at random; we did not perform any cherry-picking besides choosing one global random seed. We can see that ADA yields excellent results in all cases, and with slight truncation [19, 14], virtually all of the images look convincing. Without ADA, the convergence is hampered by discriminator overfitting, leading to inferior image quality for the original StyleGAN2, especially in METFACES, AFHQ DOG, and BRECAHAD.

Figure 17 shows examples of the generated CIFAR-10 images in both unconditional and class-conditional setting (See Appendix D.1 for details on the conditional setup). Figure 18 shows qualitative results for different methods using subsets of FFHQ at $256{\times}256$ resolution. Methods that do not employ augmentation (BigGAN, StyleGAN2, and our baseline) degrade noticeably as the size of the training set decreases, generally yielding poor image quality and diversity with fewer than 30k training images. With ADA, the degradation is much more graceful, and the results remain reasonable even with a 5k training set.

Figure 19 compares our results with unconditional BigGAN [4, 25] and StyleGAN2 config F [15]. BigGAN was very unstable in our experiments: while some of the results were quite good, approximately 50% of the training runs failed to converge. StyleGAN2, on the other hand, behaved predictably, with different training runs resulting in nearly identical FID. We note that FID has a general tendency to increase as the training set gets smaller — not only because of the lower image quality, but also due to inherent bias in FID itself [2]. In our experiments, we minimize the impact of this bias by always computing FID between 50k generated images and all available real images, regardless of which subset was used for training. To estimate the magnitude of bias in FID, we simulate a hypothetical generator that replicates the training set as-is, and compute the average FID over 100 random trials with different subsets of training data; the standard deviation was $\leq$2% in all cases. We can see that the bias remains negligible with $\geq$20k training images but starts to dominate with $\leq$2k. Interestingly, ADA reaches the same FID as the best-case generator with FFHQ-1k, indicating that FID is no longer able to differentiate between the two in this case.

Figure 20 shows additional examples of bCR leaking to generated images and compares bCR with dataset augmentation. In particular, rotations in range $[-45°, +45°]$ (denoted $\pm45°$) serve as a very clear example that attempting to make the discriminator blind to certain transformations opens up the possibility for the generator to produce similarly transformed images with no penalty. In applications where such leaks are acceptable, one can employ either bCR or dataset augmentation — we find that it is difficult to predict which method is better. For example, with translation augmentations bCR was significantly better than dataset augmentation, whereas $x$-flip was much more effective when implemented as a dataset augmentation.

Finally, Figure 21 shows an extended version of Figure 4, illustrating the effect of different augmentation categories with increasing augmentation probability $p$. Blit + Geom + Color yielded the best results with a 2k training set and remained competitive with larger training sets as well.

ADA (Ours), truncated ($\psi = 0.7$)                    Real images from the training set

ADA (Ours), untruncated                    Original StyleGAN2 config F, untruncated

FID **15.34** – KID **0.81**×10³ – Recall 0.261                    FID 19.47 – KID 3.16×10³ – Recall **0.350**

Figure 12: Uncurated 1024×1024 results generated for METFACES (1336 images) with and without ADA, along with real images from the training set. Both generators were trained using transfer learning, starting from the pre-trained StyleGAN2 for FFHQ. We recommend zooming in.

ADA (Ours), truncated ($\psi = 0.7$)

Real images from the training set

ADA (Ours), untruncated

Original StyleGAN2 config F, untruncated

FID **15.71** – KID **2.88**$\times 10^3$ – Recall **0.340**

FID 97.72 – KID 89.76$\times 10^3$ – Recall 0.027

Figure 13: Uncurated $512 \times 512$ results generated for BRECAHAD [1] (1944 images) with and without ADA, along with real images from the training set. Both generators were trained from scratch. We recommend zooming in to inspect the image quality in detail.

ADA (Ours), truncated ($\psi = 0.7$)

Real images from the training set

ADA (Ours), untruncated

Original StyleGAN2 config F, untruncated

FID **3.55** – KID **0.66**$\times 10^3$ – Recall **0.430**

FID 5.13 – KID 1.54$\times 10^3$ – Recall 0.215

Figure 14: Uncurated $512 \times 512$ results generated for AFHQ CAT [6] (5153 images) with and without ADA, along with real images from the training set. Both generators were trained from scratch. We recommend zooming in to inspect the image quality in detail.

ADA (Ours), truncated ($\psi = 0.7$)

Real images from the training set

ADA (Ours), untruncated

Original StyleGAN2 config F, untruncated

FID **7.40** – KID **1.16**$\times 10^3$ – Recall **0.454**

FID 19.37 – KID 9.62$\times 10^3$ – Recall 0.196

Figure 15: Uncurated $512 \times 512$ results generated for AFHQ Dog [6] (4739 images) with and without ADA, along with real images from the training set. Both generators were trained from scratch. We recommend zooming in to inspect the image quality in detail.

ADA (Ours), truncated ($\psi = 0.7$)

Real images from the training set

ADA (Ours), untruncated

Original StyleGAN2 config F, untruncated

FID **3.05** – KID **0.45**$\times 10^3$ – Recall **0.147**

FID 3.48 – KID 0.77$\times 10^3$ – Recall 0.143

Figure 16: Uncurated 512×512 results generated for AFHQ WILD [6] (4738 images) with and without ADA, along with real images from the training set. Both generators were trained from scratch. We recommend zooming in to inspect the image quality in detail.

Generator with best FID | Real images | Generator with best IS

Unconditional

FID **2.85** – IS 9.74   |   IS 11.24   |   FID 5.70 – IS **10.08**

Plane
Car
Bird
Cat
Deer
Dog
Frog
Horse
Ship
Truck

FID **2.38** – IS 10.00   |   IS 11.24   |   FID 3.62 – IS **10.33**

Figure 17: Generated and real images for CIFAR-10 in the unconditional setting (top) and each class in the conditional setting (bottom). We show the results for the best generators trained in the context of Figure 11b, selected according to either FID or IS. The numbers refer to the single best model and are therefore slightly better than the averages quoted in the result table. It can be seen that the model with the lowest FID produces images with a wider variation in coloring and poses compared to the model with highest IS. This is in line with the common approximation (e.g., [4]) that FID roughly corresponds to Recall and IS to Precision, two independent aspects of result quality [24, 17].

Figure 18: Images generated for different subsets of FFHQ at 256×256 resolution using the training setups from Figures 7 and 19. We show the best snapshot of the best training run for each case, selected according to FID, so the numbers are slightly better than the medians reported in Figure 7c. In addition to FID, we also report the Recall metric [17] as a more direct way to estimate image diversity. The bolded numbers indicate the lowest FID and highest Recall for each training set size. "BigGAN" corresponds to the unconditional variant of BigGAN [4] proposed by Schönfeld et al. [25], and "StyleGAN2" corresponds to config F of the official TensorFlow implementation by Karras et al. [15].

(a) Different subsets of FFHQ at 256×256

(b) Different subsets of LSUN CAT at 256×256

Figure 19: Comparison of our results with unconditional BigGAN [4, 25] and StyleGAN2 config F [15]. We report the median/min/max FID as a function of training set size, calculated over multiple independent training runs. The dashed red line illustrates the expected bias of the FID metric, computed using a hypothetical generator that outputs random images from the training set as-is.

(a) Mean images for bCR with FFHQ-5k

(b) bCR vs. dataset augment

(c) Effect of dataset $x$-flips

Figure 20: (a) Examples of bCR leaking to generated images. (b) Comparison between dataset augmentation and bCR using ±8px translations and $x$-flips. (c) In general, dataset $x$-flips can provide a significant boost to FID in cases where they are appropriate. For baseline, the effect is almost equal to doubling the size of training set, as evidenced by the consistent 2× horizontal offset between the blue curves. With ADA the effect is somewhat weaker.

Figure 21: Extended version of Figure 4, illustrating the individual and cumulative effect of different augmentation categories with increasing augmentation probability $p$.

# B Our augmentation pipeline

We designed our augmentation pipeline based on three goals. First, the entire pipeline must be strictly non-leaking (Appendix C). Second, we aim for a maximally diverse set of augmentations, inspired by the success of RandAugment [7]. Third, we strive for the highest possible image quality to reduce unintended artifacts such as aliasing. In total, our pipeline consists of 18 transformations: geometric (7), color (5), filtering (4), and corruption (2). We implement it entirely on the GPU in a differentiable fashion, with full support for batching. All parameters are sampled independently for each image.

## B.1 Geometric and color transformations

Figure 22 shows pseudocode for our geometric and color transformations, along with example images. In general, geometric transformations tend to lose high-frequency details of the input image due to uneven resampling, which may reduce the capability of the discriminator to detect pixel-level errors in the generated images. We alleviate this by introducing a dedicated sub-category, *pixel blitting*, that only copies existing pixels as-is, without blending between neighboring pixels. Furthermore, we avoid gradual image degradation from multiple consecutive transformations by collapsing all geometric transformations into a single combined operation.

The parameters for pixel blitting are selected on lines 5–15, consisting of $x$-flips (line 7), 90° rotations (line 10), and integer translations (line 13). The transformations are accumulated into a homogeneous $3 \times 3$ matrix $G$, defined so that input pixel $(x_i, y_i)$ is placed at $[x_o, y_o, 1]^T = G \cdot [x_i, y_i, 1]^T$ in the output. The origin is located at the center of the image and neighboring pixels are spaced at unit intervals. We apply each transformation with probability $p$ by sampling its parameters from uniform distribution, either discrete $\mathcal{U}\{\cdot\}$ or continuous $\mathcal{U}(\cdot)$, and updating $G$ using elementary transforms:

$$\text{SCALE2D}(s_x, s_y) = \begin{bmatrix} s_x & 0 & 0 \\ 0 & s_y & 0 \\ 0 & 0 & 1 \end{bmatrix}, \; \text{ROTATE2D}(\theta) = \begin{bmatrix} \cos\theta & -\sin\theta & 0 \\ \sin\theta & \cos\theta & 0 \\ 0 & 0 & 1 \end{bmatrix}, \; \text{TRANSLATE2D}(t_x, t_y) = \begin{bmatrix} 1 & 0 & t_x \\ 0 & 1 & t_y \\ 0 & 0 & 1 \end{bmatrix} \quad (1)$$

General geometric transformations are handled in a similar way on lines 16–32, consisting of isotropic scaling (line 17), arbitrary rotation (lines 21 and 27), anisotropic scaling (line 24), and fractional translation (line 30). Since both of the scaling transformations are multiplicative in nature, we sample their parameter, $s$, from a log-normal distribution so that $\ln s \sim \mathcal{N}\big(0, (0.2 \cdot \ln 2)^2\big)$. In practice, this can be done by first sampling $t \sim \mathcal{N}(0, 1)$ and then calculating $s = \exp_2(0.2t)$. We allow anisotropic scaling to operate in other directions besides the coordinate axes by breaking the rotation into two independent parts, one applied before the scaling (line 21) and one after it (line 27). We apply the rotations slightly less frequently than other transformations, so that the probability of applying *at least one* rotation is equal to $p$. Note that we also have two translations in our pipeline (lines 13 and 30), one applied at the beginning and one at the end. To increase the diversity of our augmentations, we use $\mathcal{U}(\cdot)$ for the former and $\mathcal{N}(\cdot)$ for the latter.

Once the parameters are settled, the combined geometric transformation is executed on lines 33–47. We avoid undesirable effects at image borders by first padding the image with reflection. The amount of padding is calculated dynamically based on $G$ so that none of the output pixels are affected by regions outside the image (line 35). We then upsample the image to a higher resolution (line 40) and transform it using bilinear interpolation (line 45). Operating at a higher resolution is necessary to reduce aliasing when the image is minified, e.g., as a result of isotropic scaling — interpolating at the original resolution would fail to correctly filter out frequencies above Nyquist in this case, no matter which interpolation filter was used. The choice of the upsampling filter requires some care, however, because we must ensure that an identity transform does not modify the image in any way (e.g., when $p = 0$). In other words, we need to use a lowpass filter $H(z)$ with cutoff $f_c = \frac{\pi}{2}$ that satisfies $\text{DOWNSAMPLE2D}\big(\text{UPSAMPLE2D}\big(Y, H(z^{-1})\big), H(z)\big) = Y$. Luckily, existing literature on wavelets [8] offers a wide selection of such filters; we choose 12-tap symlets (SYM6) to strike a balance between resampling quality and computational cost.

Finally, color transformations are applied to the resulting image on lines 48–70. The overall operation is similar to geometric transformations: we collect the parameters of each individual transformation into a homogeneous $4 \times 4$ matrix $C$ that we then apply to each pixel by computing $[r_o, g_o, b_o, 1]^T = C \cdot [r_i, g_i, b_i, 1]^T$. The transformations include adjusting brightness (line 50), contrast (line 53), and saturation (line 63), as well as flipping the luma axis while keeping the chroma unchanged (line 57) and rotating the hue axis by an arbitrary amount (line 60).

```
 1:  input: original image X, augmentation probability p
 2:  output: augmented image Y
 3:  (w, h) ← SIZE(X)
 4:  Y ← CONVERT(X, FLOAT)          ▷ Y_{x,y} ∈ [−1, +1]³

 5:  ▷ Select parameters for pixel blitting
 6:  G ← I₃       ▷ Homogeneous 2D transformation matrix
 7:  apply x-flip with probability p
 8:      sample i ∼ U{0, 1}
 9:      G ← SCALE2D(1 − 2i, 1) · G
10:  apply 90° rotations with probability p
11:      sample i ∼ U{0, 3}
12:      G ← ROTATE2D(−π/2 · i) · G
13:  apply integer translation with probability p
14:      sample t_x, t_y ∼ U(−0.125, +0.125)
15:      G ← TRANSLATE2D(round(t_x w), round(t_y h)) · G

16:  ▷ Select parameters for general geometric transformations
17:  apply isotropic scaling with probability p
18:      sample s ∼ Lognormal(0, (0.2 · ln 2)²)
19:      G ← SCALE2D(s, s) · G
20:  p_rot ← 1 − √(1 − p)       ▷ P(pre ∪ post) = p
21:  apply pre-rotation with probability p_rot
22:      sample θ ∼ U(−π, +π)
23:      G ← ROTATE2D(−θ) · G    ▷ Before anisotropic scaling
24:  apply anisotropic scaling with probability p
25:      sample s ∼ Lognormal(0, (0.2 · ln 2)²)
26:      G ← SCALE2D(s, 1/s) · G
27:  apply post-rotation with probability p_rot
28:      sample θ ∼ U(−π, +π)
29:      G ← ROTATE2D(−θ) · G    ▷ After anisotropic scaling
30:  apply fractional translation with probability p
31:      sample t_x, t_y ∼ N(0, (0.125)²)
32:      G ← TRANSLATE2D(t_x w, t_y h) · G

33:  ▷ Pad image and adjust origin
34:  H(z) ← WAVELET(SYM6)   ▷ Orthogonal lowpass filter
35:  (m_lo, m_hi) ← CALCULATEPADDING(G, w, h, H(z))
36:  Y ← PAD(Y, m_lo, m_hi, REFLECT)
37:  T ← TRANSLATE2D(½w − ½ + m_lo,x, ½h − ½ + m_lo,y)
38:  G ← T · G · T⁻¹    ▷ Place origin at image center

39:  ▷ Execute geometric transformations
40:  Y′ ← UPSAMPLE2X2(Y, H(z⁻¹))
41:  S ← SCALE2D(2, 2)
42:  G ← S · G · S⁻¹    ▷ Account for the upsampling
43:  for each pixel (x_o, y_o) ∈ Y′ do
44:      [x_i, y_i, z_i]ᵀ ← G⁻¹ · [x_o, y_o, 1]ᵀ
45:      Y_{x_o,y_o} ← BILINEARLOOKUP(Y′, x_i, y_i)
46:  Y ← DOWNSAMPLE2X2(Y, H(z))
47:  Y ← CROP(Y, m_lo, m_hi)    ▷ Undo the padding

48:  ▷ Select parameters for color transformations
49:  C ← I₄       ▷ Homogeneous 3D transformation matrix
50:  apply brightness with probability p
51:      sample b ∼ N(0, (0.2)²)
52:      C ← TRANSLATE3D(b, b, b) · C
53:  apply contrast with probability p
54:      sample c ∼ Lognormal(0, (0.5 · ln 2)²)
55:      C ← SCALE3D(c, c, c) · C
56:  v ← [1, 1, 1, 0] / √3          ▷ Luma axis
57:  apply luma flip with probability p
58:      sample i ∼ U{0, 1}
59:      C ← (I₄ − 2vᵀv · i) · C    ▷ Householder reflection
60:  apply hue rotation with probability p
61:      sample θ ∼ U(−π, +π)
62:      C ← ROTATE3D(v, θ) · C    ▷ Rotate around v
63:  apply saturation with probability p
64:      sample s ∼ Lognormal(0, (1 · ln 2)²)
65:      C ← (vᵀv + (I₄ − vᵀv) · s) · C

66:  ▷ Execute color transformations
67:  for each pixel (x, y) ∈ Y do
68:      (r_i, g_i, b_i) ← Y_{x,y}
69:      [r_o, g_o, b_o, a_o]ᵀ ← C · [r_i, g_i, b_i, 1]ᵀ
70:      Y_{x,y} ← (r_o, g_o, b_o)
71:  return Y
```

Figure 22: Pseudocode and example images for geometric and color transformations (Appendix B.1). We illustrate the effect of each individual transformation (**apply**) using four sets of parameter values, representing the 5th, 35th, 65th, and 95th percentiles of their corresponding distributions (**sample**).

1:  **input:** original image $X$, augmentation probability $p$
2:  **output:** augmented image $Y$
3:  $(w, h) \leftarrow \text{SIZE}(X)$
4:  $Y \leftarrow \text{CONVERT}(X, \text{FLOAT}) \quad \triangleright\ Y_{x,y} \in [-1, +1]^3$

5:  $\triangleright$ Select parameters for image-space filtering
6:  $b \leftarrow \left[\left[0, \frac{\pi}{8}\right], \left[\frac{\pi}{8}, \frac{\pi}{4}\right], \left[\frac{\pi}{4}, \frac{\pi}{2}\right], \left[\frac{\pi}{2}, \pi\right]\right] \quad \triangleright$ Freq. bands
7:  $g \leftarrow [1, 1, 1, 1] \qquad \triangleright$ Global gain vector (identity)
8:  $\lambda \leftarrow [10, 1, 1, 1] \,/\, 13 \quad \triangleright$ Expected power spectrum $(1/f)$
9:  **for** $i = 1, 2, 3, 4$ **do**
10:     **apply** amplification for $b_i$ **with** probability p
11:         $t \leftarrow [1, 1, 1, 1] \quad \triangleright$ Temporary gain vector
12:         **sample** $t_i \sim \text{Lognormal}\left(0, (1 \cdot \ln 2)^2\right)$
13:         $t \leftarrow t / \sqrt{\sum_j \lambda_j t_j^2} \quad \triangleright$ Normalize power
14:         $g \leftarrow g \odot t \qquad \triangleright$ Accumulate into global gain

15:  $\triangleright$ Execute image-space filtering
16:  $H(z) \leftarrow \text{WAVELET}(\text{SYM2}) \quad \triangleright$ Orthogonal 4-tap filter bank
17:  $H'(z) \leftarrow 0 \qquad\qquad \triangleright$ Combined amplification filter
18:  **for** $i = 1, 2, 3, 4$ **do**
19:     $H'(z) \leftarrow H'(z) + \text{BANDPASS}\left(H(z), b_i\right) \cdot g_i$
20:     $(m_{lo}, m_{hi}) \leftarrow \text{CALCULATEPADDING}\left(H'(z)\right)$
21:     $Y \leftarrow \text{PAD}(Y, m_{lo}, m_{hi}, \text{REFLECT})$
22:     $Y \leftarrow \text{SEPARABLECONV2D}\left(Y, H'(z)\right)$
23:     $Y \leftarrow \text{CROP}(Y, m_{lo}, m_{hi})$

24:  $\triangleright$ Additive RGB noise
25:  **apply** noise **with** probability p
26:     **sample** $\sigma \sim \text{Halfnormal}\left((0.1)^2\right)$
27:     **for each** pixel $(x, y) \in Y$ **do**
28:         **sample** $n_r, n_g, n_b \sim \mathcal{N}(0, \sigma^2)$
29:         $Y_{x,y} \leftarrow Y_{x,y} + [n_r, n_g, n_b]$

30:  $\triangleright$ Cutout
31:  **apply** cutout **with** probability p
32:     **sample** $c_x, c_y \sim \mathcal{U}(0, 1)$
33:     $r_{lo} \leftarrow \text{round}\left(\left[\left(c_x - \frac{1}{4}\right) \cdot w, \left(c_y - \frac{1}{4}\right) \cdot h\right]\right)$
34:     $r_{hi} \leftarrow \text{round}\left(\left[\left(c_x + \frac{1}{4}\right) \cdot w, \left(c_y + \frac{1}{4}\right) \cdot h\right]\right)$
35:     $Y \leftarrow Y \odot \left(1 - \text{RECTANGULARMASK}(r_{lo}, r_{hi})\right)$
36:  **return** $Y$

| Percentile: | 5th | 35th | 65th | 95th |
|---|---|---|---|---|

**Image-space filtering**

Frequency band $b_1$ $\left[0, \frac{\pi}{8}\right]$

Frequency band $b_2$ $\left[\frac{\pi}{8}, \frac{\pi}{4}\right]$

Frequency band $b_3$ $\left[\frac{\pi}{4}, \frac{\pi}{2}\right]$

Frequency band $b_4$ $\left[\frac{\pi}{2}, \pi\right]$

**Image-space corruptions**

Additive RGB noise

Cutout

Figure 23: Pseudocode and example images for image-space filtering and corruptions (Appendix B.2). $x \odot y$ denotes element-wise multiplication.

## B.2   Image-space filtering and corruptions

Figure 23 shows pseudocode for our image-space filtering and corruptions. The parameters for image-space filtering are selected on lines 5–14. The idea is to divide the frequency content of the image into 4 non-overlapping bands and amplify/weaken each band in turn via a sequence of 4 transformations, so that each transformation is applied independently with probability $p$ (lines 9–10). Frequency bands $b_2$, $b_3$, and $b_4$ correspond to the three highest octaves, respectively, while the remaining low frequencies are attributed to $b_1$ (line 6). We track the overall gain of each band using vector $g$ (line 7) that we update after each transformation (line 14). We sample the amplification factor for a given band from log-normal distribution (line 12), similar to geometric scaling, and normalize the overall gain so that the total energy is retained on expectation. For the normalization, we assume that the frequency content obeys $1/f$ power spectrum typically seen in natural images (line 8). While this assumption is not strictly true in our case, especially when some of the previous frequency bands have already been amplified, it is sufficient to keep the output pixel values within reasonable bounds.

The filtering is executed on lines 15–23. We first construct a combined amplification filter $H'(z)$ (lines 17–19) and then perform separable convolution for the image using reflection padding (lines 21–23). We use a zero-phase filter bank derived from 4-tap symlets (SYM2) [8]. Denoting the wavelet scaling filter by $H(z)$, the corresponding bandpass filters are obtained as follows (line 19):

$$\text{BANDPASS}\left(H(z), b_1\right) \;=\; H(z)H(z^{-1})H(z^2)H(z^{-2})H(z^4)H(z^{-4})/8 \qquad (2)$$

$$\text{BANDPASS}\left(H(z), b_2\right) \;=\; H(z)H(z^{-1})H(z^2)H(z^{-2})H(-z^4)H(-z^{-4})/8 \qquad (3)$$

$$\text{BANDPASS}\left(H(z), b_3\right) \;=\; H(z)H(z^{-1})H(-z^2)H(-z^{-2})/4 \qquad (4)$$

$$\text{BANDPASS}\left(H(z), b_4\right) \;=\; H(-z)H(-z^{-1})/2 \qquad (5)$$

Finally, we apply additive RGB noise on lines 24–29 and cutout on lines 30–35. We vary the strength of the noise by sampling its standard deviation from half-normal distribution, i.e., $\mathcal{N}(\cdot)$ restricted to non-negative values (line 26). For cutout, we match the original implementation of DeVries and Taylor [9] by setting pixels to zero within a rectangular area of size $\left(\frac{w}{2}, \frac{h}{2}\right)$, with the center point selected from uniform distribution over the entire image.

## C  Non-leaking augmentations

The goal of GAN training is to find a generator function $G$ whose output probability distribution $\mathbf{x}$ (under suitable stochastic input) matches a given target distribution $\mathbf{y}$.

When augmenting both the dataset and the generator output, the key safety principle is that if $\mathbf{x}$ and $\mathbf{y}$ do not match, then their augmented versions must not match either. If the augmentation pipeline violates this principle, the generator is free to learn some different output distribution than the dataset, as these look identical after the augmentations – we say that the augmentations *leak*. Conversely, if the principle holds, then the only option for the generator is to learn the correct distribution: no other choice results in a post-augmentation match.

In this section, we study the conditions on the augmentation pipeline under which this holds and demonstrate the safety and caveats of various common augmentations and their compositions.

**Notation**  Throughout this section, we denote probability distributions (and their generalizations) with lowercase bold-face letters (e.g., $\mathbf{x}$), operators acting on them by calligraphic letters ($\mathcal{T}$), and variates sampled from probability distributions by upper-case letters ($X$).

### C.1  Augmentation operator

A very general model for augmentations is as follows. Assume a fixed but arbitrarily complicated non-linear and stochastic augmentation pipeline. To any image $X$, it assigns a *distribution* of augmented images, such as demonstrated in Figure 2c. This idea is captured by an *augmentation operator* $\mathcal{T}$ that maps probability distributions to probability distributions (or, informally, datasets to augmented datasets). A distribution with the lone image $X$ is the Dirac point mass $\delta_X$, which is mapped to some distribution $\mathcal{T}\delta_X$ of augmented images.[1] In general, applying $\mathcal{T}$ to an arbitrary distribution $\mathbf{x}$ yields the linear superposition $\mathcal{T}\mathbf{x}$ of such augmented distributions.

It is important to understand that $\mathcal{T}$ is different from a function $f(X; \phi)$ that actually applies the augmentation on any individual image $X$ sampled from $\mathbf{x}$ (parametrized by some $\phi$, e.g., angle in case of a rotation augmentation). It captures the *aggregate* effect of applying this function on all images in the distribution and subsumes the randomization of the function parameters. $\mathcal{T}$ is always linear and deterministic, regardless of non-linearity of the function $f$ and stochasticity of its parameters $\phi$. We will later discuss *invertibility* of $\mathcal{T}$. Here it is also critical to note that its invertibility is not equivalent with the invertibility of the function $f$ it is based on; for an example, refer to the discussion in Section 2.2.

Specifically, $\mathcal{T}$ is a *(Markov) transition operator*. Intuitively, it is an (uncountably) infinite-dimensional generalization of a Markov transition matrix (i.e. a stochastic matrix), with nonnegative entries that sum to 1 along columns. In this analogy, probability distributions upon which $\mathcal{T}$ operates are vectors, with nonnegative entries summing to 1. More generally, the distributions have a vector space structure and they can be arbitrarily linearly combined (in which case they may lose their validity as probability distributions and are viewed as arbitrary *signed measures*). Similarly, we can do algebra with the with the operators by linearly combining and composing them like matrices. Concepts such as null space and invertibility carry over to this setting, with suitable technical care. In the following, we will be somewhat informal with the measure theoretical and functional analytic details of the problem, and draw upon this analogy as appropriate.[2]

## C.2 Invertibility implies non-leaking augmentations

Within this framework, our question can be stated as follows. Given a target distribution $\mathbf{y}$ and an augmentation operator $\mathcal{T}$, we train for a generated distribution $\mathbf{x}$ such that the augmented distributions match, namely

$$\mathcal{T}\mathbf{x} = \mathcal{T}\mathbf{y}. \tag{6}$$

The desired outcome is that this equation is satisfied only by the correct target distribution, namely $\mathbf{x} = \mathbf{y}$. We say that $\mathcal{T}$ *leaks* if there exist distributions $\mathbf{x} \neq \mathbf{y}$ that satisfy the above equation, and the goal is to find conditions that guarantee the absence of leaks.

There are obviously no such leaks in classical non-augmented training, where $\mathcal{T}$ is the identity $\mathcal{I}$, whence $\mathcal{T}\mathbf{x} = \mathcal{T}\mathbf{y} \Rightarrow \mathcal{I}\mathbf{x} = \mathcal{I}\mathbf{y} \Rightarrow \mathbf{x} = \mathbf{y}$. For arbitrary augmentations, the desired outcome $\mathbf{x} = \mathbf{y}$ does always satisfy Eq. 6; however, if also other choices of $\mathbf{x}$ satisfy it, then it cannot be guaranteed that the training lands on the desired solution. A trivial example is an augmentation that maps every image to black (in other words, $\mathcal{T}\mathbf{z} = \delta_0$ for any $\mathbf{z}$). Then, $\mathcal{T}\mathbf{x} = \mathcal{T}\mathbf{y}$ does not imply that $\mathbf{x} = \mathbf{y}$, as indeed any choice of $\mathbf{x}$ produces the same set of black images that satisfies Eq. 6. In this case, it is vanishingly unlikely that the training finds the solution $\mathbf{x} = \mathbf{y}$.

More generally, assume that $\mathcal{T}$ has a non-trivial null space, namely there exists a signed measure $\mathbf{n} \neq 0$ such that $\mathcal{T}\mathbf{n} = 0$, that is, $\mathbf{n}$ is in the null space of $\mathcal{T}$. Equivalently, $\mathcal{T}$ is not invertible, because $\mathbf{n}$ cannot be recovered from $\mathcal{T}\mathbf{n}$. Then, $\mathbf{x} = \mathbf{y} + \alpha\mathbf{n}$ for any $\alpha \in \mathbb{R}$ satisfies Eq. 6. Therefore non-invertibility of $\mathcal{T}$ implies that measures in its null space may freely leak into the learned distribution (as long as the sum remains a valid probability distribution that assigns non-negative mass to all sets). Conversely, assume that some $\mathbf{x} \neq \mathbf{y}$ satisfies Eq. 6. Then $\mathcal{T}(\mathbf{x} - \mathbf{y}) = \mathcal{T}\mathbf{y} - \mathcal{T}\mathbf{y} = 0$, so $\mathbf{x} - \mathbf{y}$ is in null space of $\mathcal{T}$ and therefore $\mathcal{T}$ is not invertible.

Therefore, leaking augmentations imply non-invertibility of the augmentation operator, which conversely implies the central principle: **if the augmentation operator $\mathcal{T}$ is invertible, it does not leak.** Such a non-leaking operator further satisfies the requirements of Lemma 5.1. of Bora et al. [3], where the invertibility is shown to imply that a GAN learns the correct distribution.

The invertibility has an intuitive interpretation: the training process can implicitly "undo" the augmentations, as long as probability mass is merely shifted around and not squashed flat.

## C.3 Compositions and mixtures

We only access the operator $\mathcal{T}$ indirectly: it is implemented as a procedure, rather than a matrix-like entity whose null space we could study directly (even if we know that such a thing exists in principle). Showing invertibility for an arbitrary procedure is likely to be impossible. Rather, we adopt a *constructive* approach, and build our augmentation pipeline from combinations of simple known-safe augmentations, in a way that can be shown to not leak. This calls for two components: a set of combination rules that preserve the non-leaking guarantee, and a set of elementary augmentations that have this property. In this subsection we address the former.

By elementary linear algebra: assume $\mathcal{T}$ and $\mathcal{U}$ are invertible. Then the composition $\mathcal{T}\mathcal{U}$ is invertible, as is any finite chain of such compositions. Hence, **sequential composition of non-leaking augmentations is non-leaking**. We build our pipeline on this observation.

The other obvious combination of augmentations is obtained by probabilistic mixtures: given invertible augmentations $\mathcal{T}$ and $\mathcal{U}$, perform $\mathcal{T}$ with probability $\alpha$ and $\mathcal{U}$ with probability $1 - \alpha$. The operator corresponding to this augmentation is the "pointwise" convex blend $\alpha\mathcal{T} + (1 - \alpha)\mathcal{U}$. More generally, one can mix e.g. a continuous family of augmentations $\mathcal{T}_\phi$ with weights given by a non-negative unit-sum function $\alpha(\phi)$, as $\int \alpha(\phi)\mathcal{T}_\phi \, \mathrm{d}\phi$. Unfortunately, **stochastically choosing among a set of augmentations is *not* guaranteed to preserve the non-leaking property**, and must be analyzed case by case (which is the content of the next subsection). To see this, consider an

---

multiplication of probability distributions in this sense (as opposed to e.g. addition of random variables), unless otherwise noted.

Technically, one can consider the vector space of finite signed measures on $\mathbb{R}^N$, which is a Banach space under the Total Variation norm. Markov operators form a convex subset of linear operators acting on this space, and general linear combinations thereof form a subspace (and a subalgebra). The exact mathematical conditions under which some of the following findings apply may be intricate but have limited practical significance given the approximate nature of GAN training.

extremely simple discrete probability space with only two elements. The augmentation operator $\mathcal{T} = \left(\begin{smallmatrix} 0 & 1 \\ 1 & 0 \end{smallmatrix}\right)$ flips the elements. Mixed with probability $\alpha = \frac{1}{2}$ with the identity augmentation $\mathcal{I}$ (which keeps the distribution unchanged), we obtain the augmentation $\frac{1}{2}\mathcal{T} + \frac{1}{2}\mathcal{I} = \frac{1}{2}\left(\begin{smallmatrix} 1 & 1 \\ 1 & 1 \end{smallmatrix}\right)$ which is a singular matrix and therefore not invertible. Intuitively, this operator smears any probability distribution into a degenerate equidistribution, from which the original can no longer be recovered. Similar considerations carry over to arbitrarily complicated linear operators.

## C.4 Non-leaking elementary augmentations

In the following, we construct several examples of relatively large classes of elementary augmentations that do not leak and can therefore be used to form a chain of augmentations. Importantly, most of these classes are not inherently safe, as they are stochastic mixtures of even simpler augmentations, as discussed above. However, in many cases we can show that the degenerate situation only arises with specific choices of mixture distribution, which we can then avoid.

Specifically, for every type of augmentation, we identify a configuration where applying it with probability strictly less than 1 results in an invertible transformation. From the standpoint of this analysis, we interpret this stochastic skipping as modifying the augmentation operator itself, in a way that boosts the probability of leaving the input unchanged and reduces the probability of other outcomes.

### C.4.1 Deterministic mappings

The simplest form of augmentation is a deterministic mapping, where the operator $\mathcal{T}_f$ assigns to every image $X$ a unique image $f(X)$. In the most general setting $f$ is any measurable function and $\mathcal{T}_f\mathbf{x}$ is the corresponding pushforward measure. When $f$ is a diffeomorphism, $\mathcal{T}_f$ acts by the usual change of variables formula with a density correction by a Jacobian determinant. These mappings are invertible as long as $f$ itself is invertible. Conversely, if $f$ is not invertible, then neither is $\mathcal{T}_f$.

Here it may be instructive to highlight the difference between $f$ and $\mathcal{T}_f$. The former transforms the underlying space on which the probability distributions live – for example, if we are dealing with images of just two pixels (with continuous and unconstrained values), $f$ is a nonlinear "warp" of the two-dimensional plane. In contrast, $\mathcal{T}_f$ operates on distributions defined on this space – think of a continuous 2-dimensional function (density) on the aforementioned plane. The action of $\mathcal{T}_f$ is to move the density around according to $f$, while compensating for thinning and concentration of the mass due to stretching. As long as $f$ maps every distinct point to a distinct point, this warp can be reversed.

An important special case is that where $f$ is a linear transformation of the space. Then the invertibility of $\mathcal{T}_f$ becomes a simpler question of the invertibility of a finite-dimensional matrix that represents $f$.

Note that when an invertible deterministic transformation is skipped probabilistically, the determinism is lost, and very specific choices of transformation could result in non-invertibility (see e.g. the example of flipping above). We only use deterministic mappings as building blocks of other augmentations, and never apply them in isolation with stochastic skipping.

### C.4.2 Transformation group augmentations

Many commonly used augmentations are built from transformations that act as a *group* under sequential composition. Examples of this are flips, translations, rotations, scalings, shears, and many color and intensity transformations. We show that a stochastic mixture of transformations within a finitely generated abelian group is non-leaking as long as the mixture weights are chosen from a non-degenerate distribution.

As an example, the four deterministic augmentations $\{\mathcal{R}_0, \mathcal{R}_{90}, \mathcal{R}_{180}, \mathcal{R}_{270}\}$ that rotate the images to every one of the 90-degree increment orientations constitute a group. This is seen by checking that the set satisfies the axiomatic definition of a group. Specifically, the set is *closed*, as composing two of elements always results in an element of the same set, e.g. $\mathcal{R}_{270}\mathcal{R}_{180} = \mathcal{R}_{90}$. It is also obviously associative, and has an identity element $\mathcal{R}_0 = \mathcal{I}$. Finally, every element has an inverse, e.g. $\mathcal{R}_{90}^{-1} = \mathcal{R}_{270}$. We can now simply speak of powers of the single generator element, whereby the four group elements are written as $\{\mathcal{R}_{90}^0, \mathcal{R}_{90}^1, \mathcal{R}_{90}^2, \mathcal{R}_{90}^3\}$ and further (as well as negative) powers "wrap over" to the same elements. This group is isomorphic to $\mathbb{Z}_4$, the additive group of integers modulo $4$.

A group of rotations is *compact* due to the wrap-over effect. An example of a *non-compact* group is that of translations (with non-periodic boundary conditions): compositions of translations are still translations, but one cannot wrap over. Furthermore, more than one generator element can be present (e.g. y-translation in addition to x-translation), but we require that these commute, i.e. the order of applying the transformations must not matter (in which case the group is called *abelian*).

Similar considerations extend to continuous *Lie groups*, e.g. that of rotations by any angle; here the generating element is replaced by an infinitesimal generator from the corresponding *Lie algebra*, and the discrete powers by the continuous exponential mapping. For example, continuous rotation transformations are isomorphic to the group $SO(2)$, or $U(1)$.

In the following subsections show that **for finitely generated abelian groups whose identity element matches the identity augmentation, stochastic mixtures of augmentations within the group are invertible, as long as the appropriate Fourier transform of the probability distribution over the elements has no zeros.**

**Discrete compact one-parameter groups**    We demonstrate the key points in detail with the simple but relevant case of a discrete compact one-parameter group and generalize later. Let $\mathcal{G}$ be a deterministic augmentation that generates the finite cyclic group $\{\mathcal{G}^i\}_{i=0}^{N-1}$ of order $N$ (e.g. the four 90-degree rotations above), such that the element $\mathcal{G}^0$ is the identity mapping that leaves its input unchanged.

Consider a stochastic augmentation $\mathcal{T}$ that randomly applies an element of the group, with the probability of choosing each element given by the probability vector $p \in \mathbb{R}^N$ (where $p$ is nonnegative and sums to 1):

$$\mathcal{T} = \sum_{i=0}^{N-1} p_i \mathcal{G}^i \tag{7}$$

To show the conditions for invertibility of $\mathcal{T}$, we build an operator $\mathcal{U}$ that explicitly inverts $\mathcal{T}$, namely $\mathcal{U}\mathcal{T} = \mathcal{I} = \mathcal{G}^0$. Whenever this is possible, $\mathcal{T}$ is invertible and non-leaking. We build $\mathcal{U}$ from the same group elements with a different weighting[3] vector $q \in \mathbb{R}^N$:

$$\mathcal{U} = \sum_{j=0}^{N-1} q_j \mathcal{G}^j \tag{8}$$

We now seek a vector $q$ for which $\mathcal{U}\mathcal{T} = \mathcal{I}$, that is, for which $\mathcal{U}$ is the desired inverse. Now,

$$\mathcal{U}\mathcal{T} = \left( \sum_{i=0}^{N-1} p_i \mathcal{G}^i \right) \left( \sum_{j=0}^{N-1} q_j \mathcal{G}^j \right) \tag{9}$$

$$= \sum_{i,j=0}^{N-1} p_i q_j \mathcal{G}^{i+j} \tag{10}$$

The powers of the group operation, as well as the indices of the weight vectors, are taken as modulo $N$ due to the cyclic wrap-over of the group element. Collecting the terms that correspond to each $\mathcal{G}^k$ in this range and changing the indexing accordingly, we arrive at:

$$= \sum_{k=0}^{N-1} \left[ \sum_{l=0}^{N-1} p_l q_{k-l} \right] \mathcal{G}^k \tag{11}$$

$$= \sum_{k=0}^{N-1} [p \otimes q]_k \mathcal{G}^k \tag{12}$$

where we observe that the multiplier in front of each $\mathcal{G}^k$ is given by the cyclic convolution of the elements of the vectors $p$ and $q$. This can be written as a pointwise product in terms of the Discrete Fourier Transform $\mathbf{F}$, denoting the DFT's of $p$ and $q$ by a hat:

$$= \sum_{k=0}^{N-1} [\mathbf{F}^{-1}(\hat{p} \odot \hat{q})]_k \mathcal{G}^k \tag{13}$$

To recover the sought after inverse, assuming every element of $\hat{p}$ is nonzero, we set $\hat{q}_i = \frac{1}{\hat{p}_i}$ for all $i$:

$$= \sum_{k=0}^{N-1} [\mathbf{F}^{-1}(\hat{p} \odot \hat{p}^{-1})]_k \mathcal{G}^k \tag{14}$$

$$= \sum_{k=0}^{N-1} [\mathbf{F}^{-1}\mathbf{1}]_k \mathcal{G}^k \tag{15}$$

$$= \mathcal{G}^0 \tag{16}$$

$$= \mathcal{I} \tag{17}$$

Here, we take advantage of the fact that the inverse DFT of a constant vector of ones is the vector $[1, 0, ..., 0]$.

In summary, the product of $\mathcal{U}$ and $\mathcal{T}$ effectively computes a convolution between their respective group element weights. This convolution assigns all of the weight to the identity element precisely when one has $\hat{q}_i = \frac{1}{\hat{p}_i}$, for all $i$, whereby $\mathcal{U}$ is the inverse of $\mathcal{T}$. This inverse only exists when the Fourier transform $\hat{p}_i$ of the augmentation probability weights has no zeros.

The intuition is that the mixture of group transformations "smears" probability mass among the different transformed versions of the distribution. Analogously to classical deconvolution, this smearing can be undone ("deconvolved") as long as the convolution does not destroy any frequencies by scaling them to zero.

Some noteworthy consequences of this are:

- Assume $p$ is a constant vector $\frac{1}{N}\mathbf{1}$, that is, the augmentation applies the group elements with uniform probability. In this case $\hat{p} = \delta_0$ and convolution with any zero-mean weight vector is zero. This case is almost certain to cause leaks of the group elements themselves. To see this directly, the mixed augmentation operator is now $\mathcal{T} := \frac{1}{N}\sum_{j=0}^{N-1}\mathcal{G}^j$. Consider the true distribution of training samples $\mathbf{y}$, and a version $\mathbf{y}' = \mathcal{G}^k\mathbf{y}$ into which some element of the transformation group has leaked. Now,

$$\mathcal{T}\mathbf{y}' = \mathcal{T}(\mathcal{G}^k\mathbf{y}) = \frac{1}{N}\sum_{j=0}^{N-1}\mathcal{G}^j\mathcal{G}^k\mathbf{y} = \frac{1}{N}\sum_{j=0}^{N-1}\mathcal{G}^{j+k}\mathbf{y} = \frac{1}{N}\sum_{j=0}^{N-1}\mathcal{G}^j\mathbf{y} = \mathcal{T}\mathbf{y} \tag{18}$$

 (recalling the modulo arithmetic in the group powers). By Eq. 6, this is a leak, and the training may equally well learn the distribution $\mathcal{G}^k\mathbf{y}$ rather than $\mathbf{y}$. By the same reasoning, any mixture of transformed elements may be learned (possibly even a different one for each image).

- Similarly, if $p$ is periodic (with period that is some integer factor of $N$, other than $N$ itself), the Fourier transform is a sparse sequence of spikes separated by zeros. Another viewpoint to this is that the group has a subgroup, whose elements are chosen uniformly. Similar to above, this is almost certain to cause leaks with elements of that subgroup.

- With more sporadic zero patterns, the leaks can be seen as "conditional": while the augmentation operator has a null space, it is not generally possible to write an equivalent of Eq. 18 without setting conditions on the distribution $\mathbf{y}$ itself. In these cases, leaks only occur for specific kinds of distributions, e.g., when a sufficient amount of group symmetry is already present in the distribution itself.

 For example, consider a dataset where all four 90 degree orientations of any image are equally likely, and an augmentation that performs either a 0 or 90 degree rotation at equal probability. This corresponds to the probability vector $p = [0.5, \ 0.5, \ 0, 0]$ over the four

elements of the 90-degree rotation group. This distribution has a single zero in its Fourier transform. The associated leak might manifest as the generator only learning to produce images in orientations 0 and 180 degrees, and relying on the augmentation to fill the gaps.

Such a leak could not happen in e.g. a dataset depicting upright faces, and the failure of invertibility would be harmless in this case. However, this may no longer hold when the augmentation is a part of a composed pipeline, as other augmentations may have introduced partial invariances that were not present in the original data.

In our augmentations involving compact groups (**rotations and flips**), we always choose the elements with a uniform probability, but importantly, only perform the augmentation with some probability less than one. This combination can be viewed as increasing the probability of choosing the group identity element. The probability vector $p$ is then constant, except for having a higher value at $p_0$; the Fourier transform of such a vector has no zeros.

**Non-compact discrete one-parameter groups**   The above reasoning can be extended to groups which are not compact, in particular **translations by integer offsets** (without periodic boundaries). In the discrete case, such a group is necessarily isomorphic to the additive group $\mathbb{Z}$ of all integers, and no modulo integer arithmetic is performed. The mixture density is then a two-sided sequence $\{p_i\}$ with $i \in \mathbb{Z}$, and the appropriate Fourier transform maps this to a periodic function. By an analogous reasoning with the previous subsection, the invertibility holds as long as this spectrum has no zeros.

**Continuous one-parameter groups**   With suitable technical care, these arguments can be extended to continuous groups with elements $\mathcal{G}_\phi$ indexed by a continuous parameter $\phi$. In the compact case (e.g. **continuous rotation**), the group elements wrap over at some period $L$, such that $\mathcal{G}_{\phi+L} = \mathcal{G}_\phi$. In the non-compact case (e.g. **translation (addition) and scaling (multiplication) by real-valued amounts**) no such wrap-over occurs. The compact and non-compact groups are isomorphic to $\mathrm{U}(1)$, and the additive group $\mathbb{R}$, respectively. Stochastic mixtures of these group elements are expressed by probability density functions $p(\phi)$, with $\phi \in [0, L)$ if the group is compact, and $\phi \in \mathbb{R}$ otherwise. The Fourier transforms are replaced by the appropriate generalizations, and the invertibility holds when the spectrum has no zeros.

Here it is important to use the correct parametrization of the group. Note that one could in principle parametrize e.g. rotations in arbitrary ways, and it may seem ambiguous as to what parametrization to use, which would appear to render concepts like uniform distribution meaningless. The issue arises when replacing the sums in the earlier formulas with integrals, whereby one needs to choose a measure of integration. These findings apply specifically to the natural *Haar measure* and the associated parametrization – essentially, the measure that accumulates at constant rate when taking small steps in the group by applying the infinitesimal generator. For rotation groups, the usual "area" measure over the angular parametrization coincides with the Haar measure, and therefore e.g. uniform distribution is taken to mean that all angles are chosen equally likely. For translation, the natural Euclidian distance is the correct parametrization. For other groups, such as scaling, the choice is a bit more nuanced: when composing scaling operations, the scale factor combines by multiplication instead of addition, so the natural parametrization is the *logarithm* of the scale factor.

For continuous compact groups (rotation), we use the same scheme as in the discrete case: uniform probability mixed with identity at a probability greater than zero.

For continuous non-compact groups, the Fourier transform of the normal distribution has no zeros and results in an invertible augmentation when used to choose among the group elements. Other distributions with this property are at least the $\alpha$-stable and more generally the infinitely divisible family of distributions. When the parametrization is logarithmic, we may instead use exponentiated values from these distributions (e.g. the log-normal distribution). Finally, stochastically mixing zero-mean normal distributed variables with identity does not introduce zeros to the FT, as it merely lifts the already positive values of the spectrum.

**Multi-parameter abelian groups**   Finally, these findings generalize to groups that are products of a finite number of single-parameter groups, provided that the elements of the different groups commute

among each other (in other words, finitely generated abelian groups). An example of this is the group of 2-dimensional translations obtained by considering x- and y-translations simultaneously.[4]

The Fourier transforms are replaced with suitable multi-dimensional generalizations, and the probability distributions and their Fourier transforms obtain multidimensional domains accordingly.

**Discussion**  Invertibility is a *sufficient* condition to ensure the absence of leaks. However, it may not always be *necessary*: in the case of *non-compact* groups, a hypothesis could be made that even a technically non-invertible operator does not leak. For example, a shift augmentation with uniform distributed offset on a continuous interval is not invertible, as the Fourier transform of its density is a sinc function with periodic zeros (except at 0). This only allows for leaks of zero-mean functions whose FT is supported on this evenly spaced set of frequencies – in other words, infinitely periodic functions. Even though such functions are in the null space of the augmentation operator, they cannot be added to any density in an infinite domain without violating non-negativity, and so we may hypothesize that no leak can in fact occur. In practice, however, the near-zero spectrum values might allow for a periodic leak modulated by a wide window function to occur for very specific (and possibly contrived) data distributions.

In contrast, straightforward examples and practical demonstrations of leaks are easily found for compact groups, e.g. with uniform or periodic rotations.

### C.4.3   Noise and image filter augmentations

We refer to Theorem 5.3. of Bora et al. [3], where it is shown that in a setting effectively identical to ours, **addition of noise that is independent of the image is an invertible operation as long as the Fourier spectrum of the noise distribution does not contain zeros**. The reason is that addition of mutually independent random variables results in a convolution of their probability distributions. Similar to groups, this is a multiplication in the Fourier domain, and the zeros correspond to irrevocable loss of information, making the inversion impossible. The inverse can be realized by "deconvolution", or division in the Fourier domain.

A potential source of confusion is that the Fourier transform is commonly used to describe spatial correlations of noise in signal processing. We refer to a different concept, namely the Fourier transform of the probability density of the noise, often called the *characteristic function* in probability literature (although correlated noise is also subsumed by this analysis).

**Gaussian product noise**  In our setting, we also randomize the magnitude parameter of the noise, in effect stochastically mixing between different noise distributions. The above analysis subsumes this case, as the mixture is also a random noise, with a density that is a weighted blend between the densities of the base noises. However, the noise is no longer independent across points, so its joint distribution is no longer separable to a product of marginals, and one must consider the joint Fourier transform in full dimension.

Specifically, we draw the per-pixel noise from a normal distribution and modulate this entire noise field by a multiplication with a single (half-)normal random number. The resulting distribution has an everywhere nonzero Fourier transform and hence is invertible. To see this, first consider two standard normal distributed random scalars $X$ and $Y$, and their product $Z = XY$ (taken in the sense of multiplying the random variables, not the densities). Then $Z$ is distributed according to the density $p_Z(Z) = \frac{K_0(|Z|)}{\pi}$, where $K_0$ is a modified Bessel function, and has the characteristic function (Fourier transform) $\hat{p}_Z(\omega) = \frac{1}{\sqrt{\omega^2 + 1}}$, which is everywhere positive [27].

Then, considering our situation with a product of a normal distributed scalar $X$ and an independent normal distributed vector $\mathbf{Y} \in \mathbb{R}^N$, the $N$ entries of the product $\mathbf{Z} = X\mathbf{Y}$ become mutually dependent. The *marginal* distribution of each entry is nevertheless exactly the above product distribution $p_Z$. By Fourier slice theorem, all one-dimensional slices through the main axes of the characteristic function of $\mathbf{Z}$ must then coincide with the characteristic function $\hat{p}_Z$ of this marginal

distribution. Finally, because the joint distribution is radially symmetric, so is the characteristic function, and this must apply to *all* slices through the origin, yielding the everywhere positive Fourier transform $\hat{p}_{\mathbf{Z}}(\boldsymbol{\omega}) = \frac{1}{\sqrt{|\boldsymbol{\omega}|^2+1}}$. When stochastically mixed with identity (as is our random skipping procedure), the Fourier Transform values are merely lifted towards 1 and no new zero-crossings are introduced.

**Additive noise in transformed bases**    Similar notes apply to additive noise in a different basis: one can consider the noise augmentation as being flanked by an invertible deterministic (possibly also nonlinear) basis transformation and its inverse. It then suffices to show that the additive noise has a non-zero spectrum in isolation. In particular, multiplicative noise with a non-negative distribution can be viewed as additive noise in logarithmic space and is invertible if the logarithmic version of the noise distribution has no zeros in its Fourier transform. The **image-space filters** are a combination of a linear basis transformation to the wavelet basis, and additive Gaussian noise under a non-linear logarithmic transformation.

### C.4.4    Random projection augmentations

The **cutout augmentation** (as well as e.g. the pixel and patch blocking in AmbientGAN [3]) can be interpreted as projecting a random subset of the dimensions to zero.

Let $\mathcal{P}_1, \mathcal{P}_2, ..., \mathcal{P}_N$ be a set of deterministic projection augmentation operators with the defining property that $\mathcal{P}_j^2 = \mathcal{P}_j$. For example, each one of these operators can set a different fixed rectangular region to zero. Clearly the individual projections have a null space (unless they are the identity projection) and they are not invertible in isolation.

Consider a stochastic augmentation that randomly applies one of these projections, or the identity. Let $p_0$, $p_1$, ..., $p_N$ denote the discrete probabilities of choosing the identity operator $\mathcal{I}$ for $p_0$, and $\mathcal{P}_k$ for the remaining $p_k$. Define the mixture of the projections as:

$$\mathcal{T} = p_0 \mathcal{I} + \sum_{j=1}^{N} p_j \mathcal{P}_j \tag{19}$$

Again, $\mathcal{T}$ is a mixture of operators, however unlike in earlier examples, some (but not all) of the operators are non-invertible. Under what conditions on the probability distribution $p$ is $\mathcal{T}$ invertible?

Assume that $\mathcal{T}$ is not invertible, i.e. there exists a probability distribution $\mathbf{x} \neq 0$ such that $\mathcal{T}\mathbf{x} = 0$. Then

$$0 = \mathcal{T}\mathbf{x} = p_0 \mathbf{x} + \sum_{j=1}^{N} p_j \mathcal{P}_j \mathbf{x} \tag{20}$$

and rearranging,

$$\sum_{j=1}^{N} p_j \mathcal{P}_j \mathbf{x} = -p_0 \mathbf{x} \tag{21}$$

Under reasonable technical assumptions (e.g. discreteness of the pixel intensity values, such as justified in Theorem 5.4. of Bora et al. [3]), we can consider the inner product of both sides of this equation with $\mathbf{x}$:

$$\sum_{j=1}^{N} p_j \langle \mathbf{x}, \mathcal{P}_j \mathbf{x} \rangle = -p_0 \langle \mathbf{x}, \mathbf{x} \rangle \tag{22}$$

The right side of this equation is strictly negative if the probability $p_0$ of identity is greater than zero, as $\mathbf{x} \neq 0$. The left side is a non-negative sum of non-negative terms, as the inner product of a vector with its projection is non-negative. Therefore, the assumption leads to a contradiction unless $p_0 = 0$; conversely, **random projection augmentation does not leak if there is a non-zero probability that it produces the identity.**

## C.5 Practical considerations

### C.5.1 Conditioning

In practical numerical computation, an operator that is technically invertible may nevertheless be so close to a non-invertible configuration that inversion fails in practice. Assuming a finite state space, this notion is captured by the condition number, which is infinite when the matrix is singular, and large when it is singular for all practical purposes. The same consideration applies to infinite state spaces, but the appropriate technical notion of conditioning is less clear.

The practical value of the analysis in this section is in identifying the conditions where exact non-invertibility happens, so that appropriate safety margin can be kept. We achieve this by regulating the probability $p$ of performing a given augmentation, and keeping it at a safe distance from $p = 1$ which for many of the augmentations corresponds to a non-invertible condition (e.g. uniform distribution over compact group elements).

For example, consider applying transformations from a finite group with a uniform probability distribution, where the augmentation is applied with probability $p$. In a finite state space, a matrix corresponding to this augmentation has $1 - p$ for its smallest singular value, and 1 for the largest, resulting in condition number $1/(1 - p)$ which approaches infinity as $p$ approaches one.

### C.5.2 Pixel-level effects and boundaries

When dealing with images represented on finite pixel grids, naive practical implementations of some of the group transformations do not strictly speaking form groups. For example, a composition of two continuous rotations of an image with angles $\phi$ and $\theta$ does not generally reproduce the same image as a single rotation by angle $\phi + \theta$, if the transformed image is resampled to the rectangular pixel grid twice. Furthermore, parts of the image may fall outside the boundaries of the grid, whereby their values are lost and cannot be restored even if a reverse transformation is made afterwards, unless special care is taken. These effects may become significant when multiple transformations are composed.

In our implementation, we mitigate these issues as much as possible by accumulating the chain of transformations into a matrix and a vector representing the total affine transformation implemented by all the grouped augmentations, and only then applying it on the image. This is possible because all the augmentations we use are affine transformations in the image (or color) space. Furthermore, prior to applying the geometric transformations, the images are reflection padded and scaled to double resolution (and conversely, cropped and downscaled afterwards). Effectively the image is then treated as an infinite tiling of suitably reflected finer-resolution copies of itself, and a practical target-resolution crop is only sampled at augmentation time.

## D Implementation details

We implemented our techniques on top of the StyleGAN2 official TensorFlow implementation[5]. We kept most of the details unchanged, including network architectures [15], weight demodulation [15], path length regularization [15], lazy regularization [15], style mixing regularization [14], bilinear filtering in all up/downsampling layers [14], equalized learning rate for all trainable parameters [13], minibatch standard deviation layer at the end of the discriminator [13], exponential moving average of generator weights [13], non-saturating logistic loss [11] with $R_1$ regularization [20], and Adam optimizer [16] with $\beta_1 = 0$, $\beta_2 = 0.99$, and $\epsilon = 10^{-8}$.

We ran our experiments on a computing cluster with a few dozen NVIDIA DGX-1s, each containing 8 Tesla V100 GPUs, using TensorFlow 1.14.0, PyTorch 1.1.0 (for comparison methods), CUDA 10.0, and cuDNN 7.6.3. We used the official pre-trained Inception network[6] to compute FID, KID, and Inception score.

| Parameter | StyleGAN2 config F | Our baseline | BreCaHAD, AFHQ | MetFaces | CIFAR-10 | + Tuning |
|---|---|---|---|---|---|---|
| Resolution | 1024×1024 | 256×256 | 512×512 | 1024×1024 | 32×32 | 32×32 |
| Number of GPUs | 8 | 8 | 8 | 8 | 2 | 2 |
| Training length | 25M | 25M | 25M | 25M | 100M | 100M |
| Minibatch size | 32 | 64 | 64 | 32 | 64 | 64 |
| Minibatch stddev | 4 | 8 | 8 | 4 | 32 | 32 |
| Dataset $x$-flips | ✓ / − | − | ✓ | ✓ | − | − |
| Feature maps | 1× | $\frac{1}{2}$× | 1× | 1× | 512 | 512 |
| Learning rate $\eta \times 10^3$ | 2 | 2.5 | 2.5 | 2 | 2.5 | 2.5 |
| $R_1$ regularization $\gamma$ | 10 | 1 | 0.5 | 2 | 0.01 | 0.01 |
| G moving average | 10k | 20k | 20k | 10k | 500k | 500k |
| Mixed-precision | − | ✓ | ✓ | ✓ | ✓ | ✓ |
| Mapping net depth | 8 | 8 | 8 | 8 | 8 | 2 |
| Style mixing reg. | ✓ | ✓ | ✓ | ✓ | ✓ | − |
| Path length reg. | ✓ | ✓ | ✓ | ✓ | ✓ | − |
| Resnet D | ✓ | ✓ | ✓ | ✓ | ✓ | − |

Figure 24: Hyperparameters used in each experiment.

### D.1 Hyperparameters and training configurations

Figure 24 shows the hyperparameters that we used in our experiments, as well as the original StyleGAN2 config F [15]. We performed all training runs using 8 GPUs and continued the training until the discriminator had seen a total of 25M real images, except for CIFAR-10, where we used 2 GPUs and 100M images. We used minibatch size of 64 when possible, but reverted to 32 for METFACES in order to avoid running out of GPU memory. Similar to StyleGAN2, we evaluated the minibatch standard deviation layer independently over the images processed by each GPU.

**Dataset augmentation**    We did not use dataset augmentation in any of our experiments with FFHQ, LSUN CAT, or CIFAR-10, except for the FFHQ-140k case and in Figure 20. In particular, we feel that leaky augmentations are inappropriate for CIFAR-10 given its status as a standard benchmark dataset, where dataset/leaky augmentations would unfairly inflate the results. METFACES, BRECAHAD, and AFHQ DOG are horizontally symmetric in nature, so we chose to enable dataset $x$-flips for these datasets to maximize result quality.

**Network capacity**    We follow the original StyleGAN2 configuration for high-resolution datasets ($\geq 512^2$): a layer operating on $N = w \times h$ pixels uses $\min\left(2^{16}/\sqrt{N}, 512\right)$ feature maps. With CIFAR-10 we use 512 feature maps for all layers. In the $256 \times 256$ configuration used with FFHQ and LSUN CAT, we facilitate extensive sweeps over dataset sizes by decreasing the number of feature maps to $\min\left(2^{15}/\sqrt{N}, 512\right)$.

**Learning rate and weight averaging**    We selected the optimal learning rates using grid search and found that it is generally beneficial to use the highest learning rate that does not result in training instability. We also found that larger minibatch size allows for a slightly higher learning rate. For the moving average of generator weights [13], the natural choice is to parameterize the decay rate with respect to minibatches — not individual images — so that increasing the minibatch size results in a longer decay. Furthermore, we observed that a very long moving average consistently gave the best results on CIFAR-10. To reduce startup bias, we linearly ramp up the length parameter from 0 to 500k over the first 10M images.

**R1 regularization**    Karras et al. [15] postulated that the best choice for the $R_1$ regularization weight $\gamma$ is highly dependent on the dataset. We thus performed extensive grid search for each column

[5] https://github.com/NVlabs/stylegan2
[6] http://download.tensorflow.org/models/image/imagenet/inception-2015-12-05.tgz

in Figure 24, considering $\gamma \in \{0.001, 0.002, 0.005, \ldots, 20, 50, 100\}$. Although the optimal $\gamma$ does vary wildly, from 0.01 to 10, it seems to scale almost linearly with the resolution of the dataset. In practice, we have found that a good initial guess is given by $\gamma_0 = 0.0002 \cdot N/M$, where $N = w \times h$ is the number of pixels and $M$ is the minibatch size. Nevertheless, the optimal value of $\gamma$ tends to vary depending on the dataset, so we recommend experimenting with different values in the range $\gamma \in [\gamma_0/5, \gamma_0 \cdot 5]$.

**Mixed-precision training**  We utilize the high-performance Tensor Cores available in Volta-class GPUs by employing mixed-precision FP16/FP32 training in all of our experiments (with two exceptions, discussed in Appendix D.2). We store the trainable parameters with full FP32 precision for the purposes of optimization but cast them to FP16 before evaluating $G$ and $D$. The main challenge with mixed-precision training is that the numerical range of FP16 is limited to $\sim \pm 2^{16}$, as opposed to $\sim \pm 2^{128}$ for FP32. Thus, any unexpected spikes in signal magnitude — no matter how transient — will immediately collapse the training dynamics. We found that the risk of such spikes can be reduced drastically using three tricks: first, by limiting the use of FP16 to only the 4 highest resolutions, i.e., layers for which $N_{layer} \geq N_{dataset}/(2 \times 2)^4$; second, by pre-normalizing the style vector $s$ and each row of the weight tensor $w$ before applying weight modulation and demodulation[7]; and third, by clamping the output of every convolutional layer to $\pm 2^8$, i.e., an order of magnitude wider range than is needed in practice. We observed about 60% end-to-end speedup from using FP16 and verified that the results were virtually identical to FP32 on our baseline configuration.

**CIFAR-10**  We enable class-conditional image generation on CIFAR-10 by extending the original StyleGAN2 architecture as follows. For the generator, we embed the class identifier into a 512-dimensional vector that we concatenate with the original latent code after normalizing each, i.e., $z' = \text{concat}\big(\text{norm}(z), \text{norm}(\text{embed}(c))\big)$, where $c$ is the class identifier. For the discriminator, we follow the approach of Miyato and Koyama [22] by evaluating the final discriminator output as $D(x) = \text{norm}\big(\text{embed}(c)\big) \cdot D'(x)^T$, where $D'(x)$ corresponds to the feature vector produced by the last layer of $D$. To compute FID, we generate 50k images using randomly selected class labels and compare their statistics against the 50k images from the training set. For IS, we compute the mean over 10 independent trials using 5k generated images per trial. As illustrated in Figures 11b and 24, we found that we can improve the FID considerably by disabling style mixing regularization [14], path length regularization [15], and residual connections in $D$ [15]. Note that all of these features are highly beneficial on higher-resolution datasets such as FFHQ. We find it somewhat alarming that they have precisely the opposite effect on CIFAR-10 — this suggests that some previous conclusions reached in the literature using CIFAR-10 may fail to generalize to other datasets.

## D.2  Comparison methods

We implemented the comparison methods shown in Figures 8a on top of our baseline configuration, identifying the best-performing hyperparameters for each method via extensive grid search. Furthermore, we inspected the resulting network weights and training dynamics in detail to verify correct behavior, e.g., that with the discriminator indeed learns to correctly handle the auxiliary tasks with PA-GAN and auxiliary rotations. We found zCR and WGAN-GP to be inherently incompatible with our mixed-precision training setup due to their large variation in gradient magnitudes. We thus reverted to full-precision FP32 for these methods. Similarly, we found lazy regularization to be incompatible with bCR, zCR, WGAN-GP, and auxiliary rotations. Thus, we included their corresponding loss terms directly into our main training loss, evaluated on every minibatch.

**bCR**  We implement balanced consistency regularization proposed by Zhao et al. [30] by introducing two new loss terms as shown in Figure 2a. We set $\lambda_{\text{real}} = \lambda_{\text{fake}} = 10$ and use integer translations on the range of $[-8, +8]$ pixels. In Figure 20, we also perform experiments with $x$-flips and arbitrary rotations.

**zCR**  In addition to bCR, Zhao et al. [30] also propose latent consistency regularization (zCR) to improve the diversity of the generated images. We implement zCR by perturbing each component of the latent $\mathbf{z}$ by $\sigma_{\text{noise}} = 0.1$ and encouraging the generator to maximize the $L_2$ difference between the

generated images, measured as an average over the pixels, with weight $\lambda_{\text{gen}} = 0.02$. Similarly, we encourage the discriminator to minimize the $L_2$ difference in $D(x)$ with weight $\lambda_{\text{dis}} = 0.2$.

**PA-GAN** Zhang and Khoreva [28] propose to reduce overfitting by requiring the discriminator to learn an auxiliary checksum task. This is done by providing a random bit string as additional input to $D$, requiring that the sign of the output is flipped based on the parity of bits that were set, and dynamically increasing the number of bits when overfitting is detected. We select the number of bits using our $r_t$ heuristic with target 0.95. Given the value of $p$ produced by the heuristic, we calculate the number of bits as $k = \lceil p \cdot 16 \rceil$. Similar to Zhang and Khoreva, we fade in the effect of newly added bits smoothly over the course of training. In practice, we use a fixed string of 16 bits, where the first $k - 1$ bits are sampled from Bernoulli(0.5), the $k^{\text{th}}$ bit is sampled from Bernoulli$\big(\min(p \cdot 16 - k + 1, 0.5)\big)$, and the remaining $16 - k$ bits are set to zero.

**WGAN-GP** For WGAN-GP, proposed by Gulrajani et al. [12], we reuse the existing implementation included in the StyleGAN2 codebase with $\lambda = 10$. We found WGAN-GP to be quite unstable in our baseline configuration, which necessitated us to disable mixed-precision training and lazy regularization, as well as to settle for a considerably lower learning rate $\eta = 0.0010$.

**Auxiliary rotations** Chen et al. [5] propose to improve GAN training by introducing an auxiliary rotation loss for $G$ and $D$. In addition the main training objective, the discriminator is shown real images augmented with $90°$ rotations and asked to detect their correct orientation. Similarly, the generator is encouraged to produce images whose orientation is easy for the discriminator to detect correctly. We implement this method by introducing two new loss terms that are evaluated on a $4\times$ larger minibatch, consisting of rotated versions of the images shown to the discriminator as a part of the main loss. We extend the last layer of $D$ to output 5 scalar values instead of one and interpret the last 4 components as raw logits for softmax cross-entropy loss. We weight the additional loss terms using $\alpha = 10$ for $G$, and $\beta = 5$ for $D$.

**Spectral normalization** Miyato et al. [21] propose to regularize the discriminator by explicitly enforcing an upper bound for its Lipschitz constant, and several follow-up works [29, 4, 30, 25] have found it to be beneficial. Given that spectral normalization is effectively a no-op when applied to the StyleGAN2 generator [15], we apply it only to the discriminator. We ported the original Chainer implementation[8] to TensorFlow, and applied it to the main convolution layers of $D$. We found it beneficial to not use spectral normalization with the `FromRGB` layer, residual skip connections, or the last fully-connected layer.

**Freeze-D** Mo et al. [23] propose to freeze the first $k$ layers of the discriminator to improve results with transfer learning. We tested several different choices for $k$; the best results were given by $k = 10$ in Figure 9 and by $k = 13$ in Figure 11b. In practice, this corresponds to freezing all layers operating at the 3 or 4 highest resolutions, respectively.

**BigGAN** BigGAN results in Figures 19 and 18 were run on a modified version of the original BigGAN PyTorch implementation[9]. The implementation was adapted for unconditional operation following Schönfeld et al. [25] by matching their hyperparameters, replacing class-conditional BatchNorm with self-modulation, where the BatchNorm parameters are conditioned only on the latent vector $z$, and not using class projection in the discriminator.

**Mapping network depth** For the "Shallow mapping" case in Figure 8a, we reduced the depth of the mapping network from 8 to 2. Reducing the depth further than 2 yielded consistently inferior results, confirming the usefulness of the mapping network. In general, we found depth 2 to yield slightly better results than depth 8, making it a good default choice for future work.

**Adaptive dropout** Dropout [26] is a well-known technique for combating overfitting in practically all areas of machine learning. In Figure 8a, we employ multiplicative Gaussian dropout for all layers of the discriminator, similar to the approach employed by Karras et al. [13] in the context of LSGAN

loss [18]. We adjust the standard deviation dynamically using our $r_t$ heuristic with target 0.6, so that the resulting $p$ is used directly as the value for $\sigma$.

### D.3 MetFaces dataset

We have collected a new dataset, MetFaces, by extracting images of human faces from the Metropolitan Museum of Art online collection. Dataset images were searched using terms such as 'paintings', 'watercolor' and 'oil on canvas', and downloaded via the `https://metmuseum.github.io/` API. This resulted in a set of source images that depicted paintings, drawings, and statues. Various automated heuristics, such as face detection and image quality metrics, were used to narrow down the set of images to contain only human faces. A manual selection pass over the remaining images was performed to weed out poor quality images not caught by automated filtering. Finally, faces were cropped and aligned to produce 1,336 high quality images at $1024^2$ resolution.

The whole dataset, including the unprocessed images, is available at
`https://github.com/NVlabs/metfaces-dataset`

## E Energy consumption

Computation is a core resource in any machine learning project: its availability and cost, as well as the associated energy consumption, are key factors in both choosing research directions and practical adoption. We provide a detailed breakdown for our entire project in Table 25 in terms of both GPU time and electricity consumption. We report expended computational effort as single-GPU years (Volta class GPU). We used a varying number of NVIDIA DGX-1s for different stages of the project, and converted each run to single-GPU equivalents by simply scaling by the number of GPUs used.

We followed the Green500 power measurements guidelines [10] similarly to Karras et al. [15]. The entire project consumed approximately 300 megawatt hours (MWh) of electricity. Almost half of the total energy was spent on exploration and shaping the ideas before the actual paper production started. Subsequently the majority of computation was targeted towards the extensive sweeps shown in various figures. Given that ADA does not significantly affect the cost of training a single model, e.g., training StyleGAN2 [15] with $1024 \times 1024$ FFHQ still takes approximately 0.7 MWh.

## Footnotes

[1]These distributions are *probability measures* over a non-discrete high dimensional space: for example, in our experiments with $256 \times 256$ RGB images, this space is $\mathbb{R}^{256*256*3} = \mathbb{R}^{196608}$.

[2]The addition and scalar multiplication of measures is taken to mean that for any set $S$ to which $\mathbf{x}$ and $\mathbf{y}$ assign a measure, $[\alpha\mathbf{x} + \beta\mathbf{y}](S) = \alpha\mathbf{x}(S) + \beta\mathbf{y}(S)$. When the measures are represented by density functions, this simplifies to the usual pointwise linear combination of the functions. We always mean addition and scalar

[3]Unlike with $p$, there is no requirement for $q$ to represent a nonnegative probability density that sums to 1, as we are establishing the general invertibility of $\mathcal{T}$ without regard to its probabilistic interpretation. Note that $\mathcal{U}$ is never actually constructed or evaluated when applying our method in practice, and does not need to represent an operation that can be algorithmically implemented; our interest is merely to identify the conditions for its existence.

[4]However, for example the non-abelian group of 3-dimensional rotations, $\mathrm{SO}(3)$, is *not* obtained as a product of the single-parameter "Euler angle" rotations along three axes, and therefore is not covered by the present formulation of our theory. The reason is that the three different rotations do not commute. One may of course still freely compose the three single-parameter rotation augmentations in sequence, but note that the combined effect can only induce a subset of possible probability distributions on $\mathrm{SO}(3)$.

[7]Note that our pre-normalization only affects the intermediate results; it has no effect on the final output of the convolution layer due to the subsequent post-normalization performed by weight demodulation.

[8]`https://github.com/pfnet-research/sngan_projection`

[9]`https://github.com/ajbrock/BigGAN-PyTorch`

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

| Item | Number of training runs | GPU years (Volta) | Electricity (MWh) |
|---|---|---|---|
| Early exploration | 253 | 22.65 | 52.05 |
| Paper exploration | 1116 | 36.54 | 87.39 |
| Setting up the baselines | 251 | 12.19 | 30.70 |
| Paper figures | 960 | 50.53 | 108.02 |
|    Fig.1     Baseline convergence | 21 | 1.01 | 2.27 |
|    Fig.3     Leaking behavior | 78 | 3.62 | 7.93 |
|    Fig.4     Augmentation categories | 90 | 4.45 | 9.40 |
|    Fig.5     ADA heuristics | 61 | 3.16 | 6.87 |
|    Fig.6     ADA convergence | 15 | 0.78 | 1.70 |
|    Fig.7     Training set sweeps | 174 | 10.82 | 22.70 |
|    Fig.8a   Comparison methods | 69 | 4.18 | 8.64 |
|    Fig.8b   Discriminator capacity | 144 | 7.70 | 15.93 |
|    Fig.9     Transfer learning | 40 | 0.71 | 1.67 |
|    Fig.11a  Small datasets | 30 | 1.71 | 4.15 |
|    Fig.11b  CIFAR-10 | 30 | 0.93 | 2.71 |
|    Fig.19   BigGAN comparison | 54 | 3.34 | 7.12 |
|    Fig.20   bCR leaks | 40 | 2.19 | 4.57 |
|    Fig.21   Cumulative augmentations | 114 | 5.93 | 12.36 |
| Results intentionally left out | 177 | 5.51 | 11.78 |
| Wasted due to technical issues | 255 | 3.86 | 8.39 |
| Code release | 375 | 12.49 | 26.71 |
| Total | 3387 | 143.76 | 325.06 |

Figure 25: Computational effort expenditure and electricity consumption data for this project. The unit for computation is GPU-years on a single NVIDIA V100 GPU — it would have taken approximately 135 years to execute this project using a single GPU. See the text for additional details about the computation and energy consumption estimates. *Early exploration* includes all training runs that affected our decision to start this project. *Paper exploration* includes all training runs that were done specifically for this project, but were not intended to be used in the paper as-is. *Setting up the baselines* includes all hyperparameter tuning for the baselines. *Figures* provides a per-figure breakdown, and underlines that just reproducing all the figures would require over 50 years of computation on a single GPU. *Results intentionally left out* includes additional results that were initially planned, but then left out to improve focus and clarity. *Wasted due to technical issues* includes computation wasted due to code bugs and infrastructure issues. *Code release* covers testing and benchmarking related to the public release.

[14] T. Karras, S. Laine, and T. Aila. A style-based generator architecture for generative adversarial networks. In *Proc. CVPR*, 2018.

[15] T. Karras, S. Laine, M. Aittala, J. Hellsten, J. Lehtinen, and T. Aila. Analyzing and improving the image quality of StyleGAN. In *Proc. CVPR*, 2020.

[16] D. P. Kingma and J. Ba. Adam: A method for stochastic optimization. In *Proc. ICLR*, 2015.

[17] T. Kynkäänniemi, T. Karras, S. Laine, J. Lehtinen, and T. Aila. Improved precision and recall metric for assessing generative models. In *Proc. NeurIPS*, 2019.

[18] X. Mao, Q. Li, H. Xie, R. Y. K. Lau, and Z. Wang. Least squares generative adversarial networks. In *Proc. ICCV*, 2017.

[19] M. Marchesi. Megapixel size image creation using generative adversarial networks. *CoRR*, abs/1706.00082, 2017.

[20] L. Mescheder, A. Geiger, and S. Nowozin. Which training methods for GANs do actually converge? In *Proc. ICML*, 2018.

[21] T. Miyato, T. Kataoka, M. Koyama, and Y. Yoshida. Spectral normalization for generative adversarial networks. In *Proc. ICLR*, 2018.

[22] T. Miyato and M. Koyama. cGANs with projection discriminator. In *Proc. ICLR*, 2018.

[23] S. Mo, M. Cho, and J. Shin. Freeze the discriminator: a simple baseline for fine-tuning GANs. *CoRR*, abs/2002.10964, 2020.

[24] M. S. M. Sajjadi, O. Bachem, M. Lucic, O. Bousquet, and S. Gelly. Assessing generative models via precision and recall. In *Proc. NIPS*, 2018.

[25] E. Schönfeld, B. Schiele, and A. Khoreva. A U-net based discriminator for generative adversarial networks. *CoRR*, abs/2002.12655, 2020.

[26] N. Srivastava, G. Hinton, A. Krizhevsky, I. Sutskever, and R. Salakhutdinov. Dropout: A simple way to prevent neural networks from overfitting. *Journal of Machine Learning Research*, 15:1929–1958, 2014.

[27] J. Wishart and M. S. Bartlett. The distribution of second order moment statistics in a normal system. *Mathematical Proceedings of the Cambridge Philosophical Society*, 28(4):455–459, 1932.

[28] D. Zhang and A. Khoreva. PA-GAN: Improving GAN training by progressive augmentation. In *Proc. NeurIPS*, 2019.

[29] H. Zhang, I. Goodfellow, D. Metaxas, and A. Odena. Self-attention generative adversarial networks. In *Proc. ICML*, 2019.

[30] Z. Zhao, S. Singh, H. Lee, Z. Zhang, A. Odena, and H. Zhang. Improved consistency regularization for GANs. *CoRR*, abs/2002.04724, 2020.