[Reviews · NeurIPS 2020]

Review 1

Summary and Contributions: This paper proposes a new effective method for training GANs from small size of data by incorporating non-leaking data augmentation. For achieving this, the authors design an adaptive discriminator augmentation (ADA). They extensively analyze why data augmentation can harm the GAN performance with diverse experiments. The ADA employs a data augmentation probability for generator as well as discriminator and a heuristic metric for measuring overfitting status. They evaluate ADA on various datasets such as FFHQ, LSUN Cat, AFHQ-Dog, MetFaces, BreCaHad, and CIFAR-10 comparing with SOTA methods including bCR. The results are very promising.

Strengths: - Robust GAN training from small datasets is very important and challenging. - Even if some recent studies proposed data augmentation-based GAN training, this method handles leaky data augmentation problem. - The proposed method is simple but effective. - The paper is well-written and clear. - The authors provide extensive analysis on leaky augmentation and promising results on many datasets including interpolation videos. - They will release a new MetFace dataset.

Weaknesses: Basically, I like this paper. - How about the results when not applying augmentation to G? Because the overall flow of the proposed method is different from bCR, D only augmentation results might be meaningful for supporting the hypothesis. - In Figure2, what yellowgreen boxes mean? - The minibatch numbers to adjust p is set to 4. How sensitive is the performance to the number? - What is the reason of large r_t fluctuation for fixed p in Figure 5(d)? More discussion will be helpful for readers - For AFHQ dataset, the authors presented the results of the dog domain. How is the results of the wild-life domain where the intra-domain variance is larger? Similar or not?

Correctness: The claims are clear and correct.

Clarity: The paper is well written and easy to follow.

Relation to Prior Work: Clearly discussed.

Reproducibility: Yes

Additional Feedback: - Overall, this method deals with discriminator overfitting issue with non-leaky data augmentation. It is well-know that model capacity affects the effects of data augmentation methods. Strong augmentation such as mixup [Zhang et al. 2018] and cutmix [Yun et al. 2019] make more improvements for larger models (ResNet, EfficientNet-L) than smaller ones (mobilenet_v2). Then, given larger training dataset, if the parameter size of discriminator increases, data augmentation can make some effects? Or any results on larger discriminator? Of course, I know this is beyond the scope of this paper. - In Figure 1, it will be better to add "black dots mean the best points." - In Figure 3, for color transformation, the patterns of p are different depending on each color? for example, when fixed color (blue tone, green tone, yellow tone, etc), the p patterns are different? - In Figure 7(d), quantitative values can be helpful such as pixel-wise L1 difference between two mean images. [Zhang et al. 2018] mixup: Beyond Empirical Risk Minimization. ICLR 2018. [Yun et al. 2019] CutMix: Regularization Strategy to Train Strong Classifiers with Localizable Features. ICCV 2019. After rebuttal: ========================================== I thank the authors for their great efforts. I carefully read the other reviewers' comments and author response. The authors clearly answered my questions. I decided to raise my score to 9 considering the importance of this topic.


Review 2

Summary and Contributions: This work proposes to address the problem of limited data in GAN training with discriminator augmentation (DA), a technique which enables most standard data augmentation techniques to be applied to GANs without leaking them into the learned distribution. The method is simple, yet effective: non-leaking differentiable transformations are applied to real and fake images before being passed through the discriminator, both during discriminator and generator updates. To make transformations non-leaking, it is proposed to apply them with some probability p < 1 such that the discriminator will eventually be able to discern the true underlying distribution. One challenge introduced with this technique is that different datasets require different amounts of augmentation depending on their size, and as such, expensive grid search is required for optimization. To eliminate the need for this search step an adaptive version called adaptive discriminator augmentation (ADA) is introduced. ADA monitors discriminator overfitting and adjusts the strength of the augmentation accordingly throughout training. It is shown that ADA improves image generation quality significantly in the limited data setting, outperforming all competing regularization and transfer learning methods. A new dataset called MetFaces is introduced as a high resolution, low data option. Additionally, significant improvements over current state-of-the-art results are achieved on the popular CIFAR-10 benchmark.

Strengths: S1 - Data augmentation is an extremely common tool in classification, but until now has been sorely missing from the GAN literature. This work is very likely to have widespread appeal considering the current popularity of GANs. S2 - Principled definition for what constitutes a non-leaking augmentation operator. S3 - Adaptive method eliminates the need for costly grid searches. S4 - Considerable improvement in limited data setting, which has long been a sore spot for GANs. S5 - State-of-the-art performance on the CIFAR-10 dataset by a large margin. Considering how popular this dataset is for benchmarking GAN performance, this is no small feat.

Weaknesses: None that I found noteworthy.

Correctness: Co1 - Comparison between the proposed technique and competing methods is fair. Care is taken to properly optimize each method, rather than simply using default settings.

Clarity: Cl1 - Paper is well written and easy to follow. Figures do a great job of summarizing key points. Supplementary material contains extensive details for reproducibility and further insights.

Relation to Prior Work: RPW1 - Extensive discussion and comparison with related methods.

Reproducibility: Yes

Additional Feedback: AF1 - This is fantastic work! Proper data augmentation has been missing from GANs for a long time, and this work fills that hole. Having adaptive augmentation strength as well is icing on the cake. The supplementary material is top class - very detailed and contains many insights that will be useful for those trying to reimplement or extend this work in the future. == Post Rebuttal == After reading the rebuttal and other reviewers' comments I have decided to maintain my scoring on this paper. The problem addressed is one of widespread interest, and the paper provides many details and insights that will be of use to those who would like to use or build on this work.


Review 3

Summary and Contributions: This paper investigates the overfiting problem of GANs given limited data. The overfitting appear in discriminator. This paper consider data augmentation to eliminate this issue. Authors exploit many data augmentation methods, and explore the combination of data augmentations. From experiments authors find that the performance of model is effected by training process, and thus proposed a new adaptive discriminator augmentation, which is computed on the output of discriminator. Both qualitative and quantitative results demonstrate that the proposed method achieves good results.

Strengths: Given the few data this paper exploits the overfitting problem of GANs, which is in discriminator. Pros: The paper designs one interesting experiment to investigate the output of discriminator, which evaluates the distance of the train image, generated image and validation image. I like this experiment, which considers three sets to check what happen in D. Figure 1 show the output validation image is similar to the one of the generated image, which means the D heavily remember the train image. The comprehensive analysis is conducted. Specially, it refers to the overfittng, the way of data augmentation, the probability, the dataset size, different datasets, and transfer learning. The paper is easy to follow

Weaknesses: I have a few questions: 1. If using different GAN loss (here WGAN-GP) to train, the output distribution of D (Figure c) will be similar or not. What I means is that D(real) is always larger than D(fake). What happen if we use hinge loss ? 2. In Figure 8(a), the result of spectral norm is worse. Why it happen here? I know it is weird to ask since it is not from this paper. 3. The title should be more specific and include information like 'data augmentation', which is fast to get the key point from the title. There are a few techniques to reduce the overfitting, such as the data augmentation, regularization, transfer learning etc. I would like to add specific information.

Correctness: It is vary clear

Clarity: It is good paper, and easy to follow.

Relation to Prior Work: Authors concludes the related work.

Reproducibility: Yes

Additional Feedback: --------------------- AFTER REBUTTAL --------------------- I thank authors for rebuttal. Authors address my concern. I would like to keep my score.


Review 4

Summary and Contributions: The paper proposes to learn gans given limited number of data. The idea is to perform data augmentation in the discriminator. The augmentation is carefully designed to ensure that the generator will converge to the data distribution if the augmented distributions match. The proposed method is evaluated in several visual tasks.

Strengths: I agree that learning gans in the limited data setting is important and interesting. The proposed method is simple and makes sense. The augmentation is carefully designed to ensure that the generator will converge to the data distribution if the augmented distributions match. The results on several visual tasks are impressive.

Weaknesses: The paper misses some theoretical insight of the proposed method. The discussion about the "non-leaking" augmentation operators only considers the equilibrium point, which can be hardly achieved in practice. Then how this artificial augmented data can lead to a better generator in an adversarial game is still unclear for me. The main hypothesis of the paper is that it can prevent the discriminator from overfitting the training data. Then, what if we use a smaller discriminator, early stoping or other techniques in the discriminator to prevent overfitting. Will the results be the same? An empirical comparison is necessary and further analysis of the improvements is preferable ------after rebuttal------- Thanks for the author feedback. I agree that the proposed method won't change the equilibrium. I won't fault the author for the missing analysis of the empirical convergence. Besides, the authors claim that other techniques including smaller and finetuned discriminator and early stopping won't prevent the discriminator from overfitting, which strengthens the motivation. I raise my score from 6 to 7.

Correctness: Yes.

Clarity: Yes.

Relation to Prior Work: The most related paper is discussed.

Reproducibility: Yes

Additional Feedback:

[Author Response · NeurIPS 2020]

We thank the reviewers for insightful feedback and suggestions. We will clarify our figures and text accordingly.

**Intended role of theory and experiments (R4).** The goal of our theory is to show that non-leaking augmentations
*do not inherently harm* the training objective — the results would inevitably degrade if the equilibrium point was
affected by the augmentations. However, we agree that the equilibrium is hardly ever achieved in practice, and that the
effectiveness of augmentations ultimately depends on the complex interaction between many aspects of the training
process. Thus, in order to *demonstrate the benefits* of our technique, we rely on extensive practical experiments and
place considerable emphasis on comparing against a sufficiently large set of alternative approaches, including e.g.
adaptive dropout (Figure 8a).

The reason why augmentations help in the adversarial game is that they make it harder for the raw D outputs of real and
generated images to drift apart, as visualized in Figures 1 and 6. This is important because the gradients that G receives
from D become meaningless once the overlap between the distributions is lost.

**Early stopping (R4).** As can be seen in Figure 6a, our method typically leads to monotonic convergence that clearly
surpasses the best FID achievable using early stopping (Figure 1a). In the context of GANs, it is customary to report
the lowest FID seen over the course of training and we also follow the same protocol. In this sense, our experiments
already employ early stopping — largely to the benefit of the comparison methods. With ADA, this would not be strictly
necessary since we could get comparable results by only looking at FID toward the end of the training in most cases.

**Discriminator capacity (R1, R4).** We agree with the reviewers that a sweep over D capacity would provide valuable
insight and will gladly include it in the final version. That said, we have not observed that decreasing the capacity would
prevent overfitting with small training sets ($\sim$2k), but it might reduce or postpone it slightly with moderately-sized ones
($\sim$30k). Similarly, we have not observed significant benefits from increasing the capacity, either.

**Spectral normalization (R3).** The StyleGAN2 paper also reports that spectral normalization did not help, in line with
our results in Figure 8a. We do not know exactly why this is the case but would like to emphasize that the interaction
between various regularization techniques and architectural choices is not fully understood yet. We suspect that the
effectiveness of spectral normalization is tied to a specific kind of training setup that is sufficiently different from the
one used in StyleGAN2. For example, papers where spectral normalization is shown to be beneficial do not typically
employ explicit gradient penalty terms, such as $R_1$.

**Different loss functions (R3).** The exact behavior of D certainly depends on the loss function. We chose to
focus on the original non-saturating logistic loss that was used in StyleGAN2. We suspect that hinge loss would
exhibit comparable behavior, as the shape of the two functions is substantially similar: $f(x) = \log\left(\mathrm{sigmoid}(x)\right)$ vs.
$f(x) = -max(0, 1-x)$. WGAN and WGAN-GP are somewhat trickier, because the outputs of D are not "grounded" to
any particular range, so their mean and standard deviation tend to drift around over the course of training. Nevertheless,
we would generally expect $D_{\mathrm{train}}$ to stay above $D_{\mathrm{generated}}$, with the difference becoming more pronounced when D starts
to overfit.

**D-only augmentation (R1).** Let us consider what would happen if the augmentations were only applied when training
D, but skipped when training G. In this case, D would see the true distribution of generated images ($\mathbf{x}$) when G is being
trained and the augmented distribution ($\mathcal{T}\mathbf{x}$) when D itself is being trained. We have tested this variant and observed
that the mismatch between these two distributions leads to an immediate mode collapse. In effect, D is only trained to
guide $\mathcal{T}\mathbf{x}$ toward $\mathcal{T}\mathbf{y}$, so it is unable to provide meaningful gradients for guiding $\mathbf{x}$ toward $\mathbf{y}$. The situation is markedly
different with bCR, because the main training objective of D is still based on the true distributions — the augmented
distributions are used only in the auxiliary loss terms, so their effect is weaker and less direct.

**Fluctuations in Figure 5d (R1).** We have noticed that non-adaptive discriminator augmentation tends to cause
fluctuation in the training dynamics once D has entered the overfitting regime, i.e., when FID has started increasing.
This often happens toward the end of the training when the specific choice of fixed $p$ is no longer sufficient to prevent
overfitting. We suspect that the fluctuation is caused by D having become overly sensitive to a small set of image
features and reacting strongly to the stochastic effect of the augmentations on these features.

**Clarifications (R1).** In Figure 2, the yellow boxes indicate the loss function and the green boxes indicate the network
being trained. In the ADA heuristic, we adjust $p$ every 4 minibatches simply because of the way the StyleGAN2 training
loop is laid out; the results are not sensitive to this particular choice. Regarding the AFHQ dataset, we have tested the
CAT and WILD categories and observed high-quality results comparable to the DOG category. We originally left these
results out to save space but could include them in the final version.

[Meta-Review · NeurIPS 2020]

All reviewers found this work interesting and addressing an important issue in GAN training. The authors did a great job in presenting their analyses and experiments. Please take the reviewers' comments into account in your next revision (particularly some presentation advices). The authors are encouraged to cite the following work for a similar "non-leaking" DA: https://arxiv.org/abs/2006.05338 (To be clear, the meta-reviewer has absolutely zero ties to the above work but just happens to be aware of it. We did not bring this out during discussion nor used this for or against the authors.)